

# Linking Glacier Retreat with Climate Change on the Tibetan Plateau through Satellite Remote Sensing

Fumeng Zhao[1], Wenping Gong[1], Silvia Bianchini[2], Zhongkang Yang[3]

[1]Faculty of Engineering, China University of Geosciences, Wuhan, 430074, China
5 [2]Earth Sciences Department, University of Florence, Florence, 50121, Italy
[3]PowerChina Chengdu Engineering Corporation Limited, Chengdu, 610072, China

*Correspondence to*: Wenping Gong (wenpinggong@cug.edu.cn)

**Abstract.** Under global climate change, glaciers on the Tibetan Plateau are experiencing severe retreat, which significantly impacts the regional water cycle and the occurrence of natural hazards. However, detailed insights into the spatial-temporal patterns of this retreat and its climatic drivers remain insufficiently explored. In this study, an Adaptive Glacier Extraction Index (AGEI) is proposed based on analysis of multispectral Landsat images integrated with the Google Earth Engine, and comprehensive and high-resolution mapping of glaciers on the Tibetan Plateau is realized at five-year intervals from 1988 to 2022; subsequently, the ERA5-Land air temperature and precipitation data are downscaled to a finer 1-km resolution; finally, impacts of the annual and seasonal change of downscaled meteorological factors on the glacier retreat are quantified. Results demonstrated a rapid yet heterogeneous pattern of glacier retreat across the Tibetan Plateau between 1988 and 2022, with retreat rates ranging from $0.14 \pm 0.07\%$ to $0.51 \pm 0.09\%$ annually. A notable trend was observed where most glacier areas experienced a decrease from 1990 to 2000, followed by a slight increase. From 2010, a majority of the glaciers exhibited either a static state or minimal retreat. The most pronounced impact of annual temperature on glacier retreat is observed in Zone VIII, with a value of $-9.34 \times 10^3$ km$^2$/°C, and the most restraining impact of precipitation on glacier retreat reaches 261 km$^2$/mm, which is observed in Zone VI for the spring season. These insights are pivotal in comprehending the temporal and spatial heterogeneity of glacier retreats, and in understanding the effects of climatic variations on the state of glaciers on the Tibetan Plateau.

## 1 Introduction

The Tibetan Plateau, often referred to as the Third Pole, hosts the largest concentration of glaciers outside the polar regions 25 (Yao et al., 2012). These glaciers are integral to the regional water cycle and are a vital source of water resources for downstream areas (Zhang et al., 2021; Salerno et al., 2023). However, the current glacier mass balance has indicated severe glacier retreat (excluding the Karakoram Anomaly) at an accelerated rate primarily due to climate warming (Curio et al., 2015; Farinotti et al., 2020; Zhang et al., 2021; Chen et al., 2023). Prior research has documented that in High Mountain Asia, the glacier area experienced a reduction of $0.43 \pm 0.19\%$ annually from 1990 to 2018, and the glacial mass loss was 30 quantified at $16.3 \pm 3.5$ Gt per annum between 2000 and 2016 (Brun et al., 2017; Huang et al., 2021). This pronounced




retreat of glaciers is inducing considerable alterations in runoff patterns on seasonal, interannual, and decadal scales. Concurrently, there is an escalating risk of glacial hazards, including avalanches, glacial debris flows, and glacial lake outburst floods. These developments pose a significant threat to life and property across the region (Lamsal et al., 2016; Lin et al., 2021; Zhao et al., 2022).

Glacier changes encompass a spectrum of metrics, including variations in area, thickness, volume, surface mass balance, and equilibrium line altitude (Sugiyama et al., 2013; Su et al., 2022). These parameters for individual glaciers can be monitored through geodetic and glaciological techniques (Zemp et al., 2019). The advent and enhanced accessibility of high-resolution, high-quality satellite imagery have enabled comprehensive investigations of glacier changes on a regional scale (Che et al., 2020; Beraud et al., 2023). Over recent decades, the focus on the Tibetan Plateau has been towards conducting glacier

mapping and change analyses at the basin level, utilizing satellite remote sensing data (Wang et al., 2013; Neckel et al., 2014; Ye et al., 2017; Zhang et al., 2021). Glaciers are commonly mapped using a variety of techniques, including spectral analysis of optical satellite images, object segmentation-based methods, and supervised machine-learning algorithms (Bolch et al., 2010; Robson et al., 2015; Khan et al., 2020). Additionally, Synthetic Aperture Radar (SAR) coherence images are increasingly utilized for glacier mapping, leveraging their ability to penetrate cloud cover (Holobâcă et al., 2021). Among

these methods, robust and efficient pixel-based multispectral analysis has been particularly effective in accurately delineating debris-free glaciers. This approach harnesses the distinct spectral signatures of ice and glaciers, allowing for precise identification and mapping of glacier boundaries (Huang et al., 2021; Bevington and Menounos, 2022). However, the magnitude of glacier retreat on the Tibetan Plateau varies significantly, influenced by marked spatiotemporal variations in climate conditions and topographical factors (Ye et al., 2017; Latif et al., 2019). Furthermore, the endeavor of large-scale

mapping of glacier retreat is particularly challenging, primarily due to the scarcity of satellite observations that possess both the necessary quality and resolution. As a result, the comprehensive depiction of glacier retreat across the entire Tibetan Plateau, especially at finer temporal and spatial scales, remains inadequately characterized (Shean et al., 2020; Xiao et al., 2023). Recent advancements in cloud computing platforms, such as Google Earth Engine, have significantly enhanced the capability for automated glacier mapping. By harnessing extensive archives of satellite imagery, the Google Earth Engine

platform facilitates a more comprehensive understanding of the impacts of global climate change on the cryosphere (Gorelick et al., 2017; Shugar et al., 2020; Huang et al., 2021; Bevington and Menounos, 2022).

Generally, regional climatic conditions determine the intensity of glacier ablation and accumulation (i.e., mass balance) over prolonged periods, and climate change is recognized to be the dominant driver of the glacier mass balance and the associated area and volume changes (Su et al., 2022). Glaciers serve as sensitive indicators of climate, primarily responding to

interannual variations in temperature and precipitation (Harrison, 2013; Shean et al., 2020). One critical challenge in understanding glacier responses to climatic changes is the significant variability in glacier sensitivities. These sensitivities and the associated feedback mechanisms can either accelerate or restrain glacier melting (Johnson and Rupper, 2020). Consequently, the availability of high-resolution meteorological data becomes essential for accurately discerning the patterns and dynamics of glacier retreats in response to climate change (Rashid and Majeed, 2018). Obtaining high-resolution gridded



meteorological data at a regional level typically involves interpolations of rain gauge observations and satellite estimates (Crespi et al., 2019; Afonso et al., 2020). However, the applicability of these methods is often limited in complex terrains, primarily due to the sparse observations and their inherently coarse spatial resolutions. Statistical downscaling techniques are routinely applied to refine the spatial resolution of temperature and precipitation datasets by leveraging ancillary variables available at finer scales (Ebrahimy et al., 2021; Jiang et al., 2021). Research has consistently demonstrated a correlation between climatic variables and a range of biophysical factors, including topography, land cover types, vegetation cover, surface albedo, and soil moisture status (Hutengs and Vohland, 2016; Zhang et al., 2019a). The integration of these high-resolution ancillary variables enables the production of more detailed meteorological products, providing a more accurate representation of local climatic conditions essential for various environmental assessments. Moreover, meteorological reanalysis datasets, such as ERA5-Land, are continually updated, thereby facilitating analyses of climate trends and anomalies. These datasets can be effectively utilized as climate inputs for downscaling analyses (Muñoz-Sabater et al., 2021; Wang et al., 2021; Kusch and Davy, 2022; Salerno et al., 2023).

Despite numerous studies examining glacier variations on the Tibetan Plateau in recent decades, the specific impacts of climate change on glacier retreat have not been thoroughly investigated at a finer resolution. This study is designed with three primary objectives: (1) to produce a high-resolution mapping of glacier retreat on the Tibetan Plateau, capturing changes at five-year intervals from 1988 to 2022; (2) to acquire downscaled meteorological products, enhancing the spatial resolution of climatic data; and (3) to quantify the impacts of annual and seasonal climate change on the patterns and rates of glacier change. The rest of this paper is organized as follows. First, the information of the study area is provided; second, the methodologies used for glacier mapping and downscaling analysis are depicted; third, the results of the time-series glacier mapping and the derived downscaled meteorological products are presented; fourth, the influence of annual and seasonal changes in meteorological factors on glacier retreat is analyzed; finally, the concluding remarks are provided.

## 2 Information of the Study Area

The Tibetan Plateau, encompassing an expansive area of approximately $3 \times 10^6$ km$^2$ and characterized by an average elevation of about 4300 m, has witnessed notable climatic shifts in recent decades (Royden et al., 2008; Zhang, 2019b). The region has experienced a pronounced warming trend, with air temperatures rising approximately 0.3 to 0.4 °C/decade over the past half-century. Concurrently, there has been a marginal yet observable increase in precipitation levels across the plateau (Bibi et al., 2018; Zhang et al., 2021). As shown in Fig. 1, the atmospheric circulation patterns on the Tibetan Plateau are predominantly governed by the interplay between the East Asian and Indian monsoon systems during summer and the influence of mid-latitude Westerlies during winter. These circulation patterns, in conjunction with distinct topography, exert a significant influence on the climatic factors that control the variability and distribution of glaciers (Yao et al., 2012). The intricate dynamics of these atmospheric systems, coupled with the unique geographical features of the plateau, play a crucial role in shaping the climatic conditions that directly impact glacier behavior and distribution (Sun et al., 2018).





To systematically evaluate the status of glaciers and comprehend the influences of climate change on their retreat, the Tibetan Plateau is divided into eight distinct sub-zones. This division primarily hinges on the patterns of large-scale atmospheric circulation, as shown in Fig. 1. In addition, the delineation of these sub-zones incorporates considerations of the drainage basins. This process is further refined using the HydroBASINS dataset (https://www.hydrosheds.org/), which aids in ensuring accurate differentiation of the sub-zones. Depicted in Fig. 1 are the delineation results of the eight sub-zones. Zones I, II, and VI are predominantly influenced by the East Asian monsoon, Indian monsoon, and Westerlies, respectively. Zone III experiences impacts from both the East Asian and Indian monsoons, while the interplay of the Westerlies and the East Asian monsoon governs Zone VII. Similarly, Zone VIII is under the influence of both the Westerlies and the Indian monsoon. Zone IV occupies the transitional area between the East Asian and Indian monsoons and Zone V is situated in the transitional zone between the Westerlies and the Indian monsoon.

This research utilizes the Randolph Glacier Inventory (RGI) 6.0 as the reference for glacier outlines (RGI Consortium, 2017). The RGI 6.0 is compiled through a combination of manual and automated digitization (Scherler et al., 2018). Fig. 1 illustrates that the majority of glaciers cataloged in the RGI 6.0 are concentrated in Zones II, VI, and VIII, predominantly situated in the Karakoram and Himalayas. The glaciers on the Tibetan Plateau, occupying diverse terrain conditions, distinctly reflect the variations in atmospheric circulation processes.

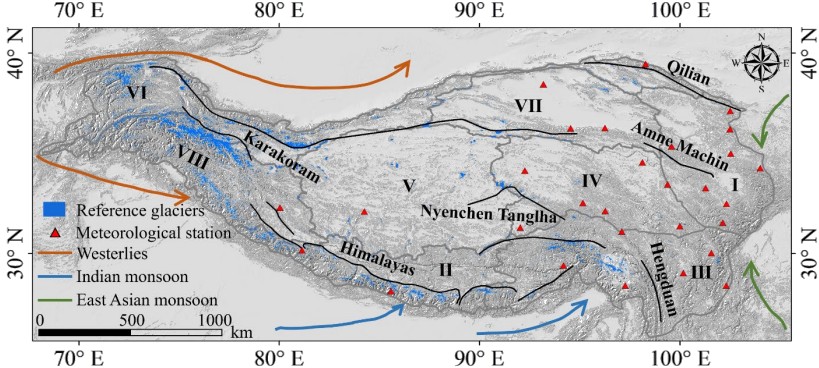

**Figure 1: Delineation results for the eight sub-zones on the Tibetan Plateau and the reference glaciers obtained from RGI 6.0. The background image is the topographic map from the Shuttle Radar Topographic Mission (SRTM) Digital elevation model (DEM) provided by the USGS EROS Archive (EROS Centre, 2018).**

## 3 Materials and Methods

This section introduces the dataset and method for time-series glacier mapping, followed by a detailed description of the downscaling analysis of air temperature and precipitation datasets to assess climate change impacts on glacier retreat.





## 3.1 Glacier mapping through time series Landsat images

The time series of multispectral satellite images utilized for glacier mapping encompasses data from the Landsat-4 and Landsat-5 Thematic Mapper (TM), Landsat-7 Enhanced Thematic Mapper Plus (ETM+), and Landsat-8 Operational Land Imager (OLI) sensors. This selection is attributed to their prolonged data availability period, comparatively high spatial resolution, and open access. These orthorectified images are processed using the Google Earth Engine platform.

The identification of debris-covered glaciers from satellite imagery poses significant challenges, as distinguishing debris from snow and ice is feasible only when glacier boundaries are predefined (Robson et al., 2015). Thus, this study focuses solely on debris-free glaciers. In this study, an adaptive glacier extraction index (AGEI) method is proposed to accomplish the time-series glacier mapping. The AGEI method is primarily based on the analysis of the Normalized Difference Snow Index (NDSI). Numerous studies have demonstrated the efficacy of the NDSI in distinguishing glacier ice from non-ice areas, particularly in shadowed regions (Burns and Nolin, 2014; Huang et al., 2021; Bevington and Menounos, 2022). Nevertheless, the potential for misidentification persists, primarily due to factors like glacial lake presence and seasonal snow cover. To reduce such errors, the AGEI method also encompasses the Normalized Difference Water Index (NDWI) and the surface temperature. The proposed AGEI method for glacier mapping is delineated in the following three steps:

**(1) Generation of Landsat data cube at five-year intervals spanning from 1988 to 2022**

Obtaining cloudless Landsat images on the Tibetan Plateau presents significant challenges, primarily due to the impact of shadows at low sun angles and the relatively short duration of the snow-free season (Liu et al., 2020). To mitigate the influence of clouds and seasonal snow on glacier mapping, this study uses the data cube from the Landsat data collection, and the acquisition dates of the atmospherically corrected Landsat images are restricted to the ablation season (from June 1 to August 31). Subsequently, the cloud score is calculated for each pixel in the data cube, ranging from 0 (indicating cloud-free) to 100 (representing very thick cloud cover). In this study, the cloud score threshold is set to 60, balancing the need for a sufficient number of image pixels against the accuracy of the mapped glaciers. In addition, the quality of Landsat images on the Tibetan Plateau is affected by cloud coverage, and the Landsat images in one year may not generate a relatively cloud-free image mosaic. Although there may be large interannual variability of glacier area, this study analyzed the glacier area changes every five-year interval to increase the number of available Landsat images and to improve the robustness of glacier area change. The acquisition dates and the corresponding number of available Landsat images used for each glacier map are detailed in Table 1. Furthermore, Fig. 2 depicts the number of Landsat images available for each year corresponding to each period of glacier mapping. It can be seen that the number of Landsat images per year corresponding to each period is relatively evenly distributed except for the period of 2010. In particular, the number of Landsat images in 2012 was 0, which can be attributed to the data availability of Landsat 5 from March 1984 to May 2012. In addition, the Landsat 8 mission was successfully launched in February 2013; thus, the number of available Landsat images has increased since 2013.





**Table 1: Acquisition dates and corresponding number of Landsat images for glacier mapping.**

| Periods | Dates | Sensors | Number of Landsat images |
|---------|-----------|-----------|--------------------------|
| 1990 | 1988 - 1992 | TM | 2004 |
| 1995 | 1993 - 1997 | TM | 2696 |
| 2000 | 1998 - 2002 | TM, ETM+ | 2223 |
| 2005 | 2002 - 2007 | TM, ETM+ | 2206 |
| 2010 | 2008 - 2012 | TM, ETM+ | 2911 |
| 2015 | 2013 - 2017 | OLI | 4568 |
| 2020 | 2017 - 2022 | OLI | 4530 |


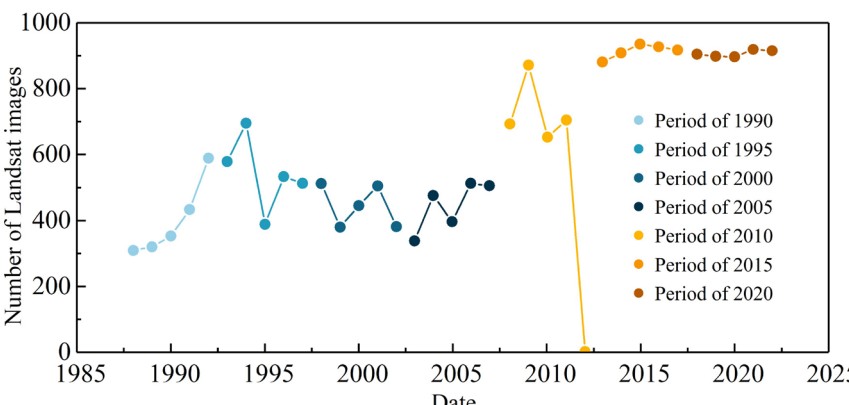

**Figure 2: Number of Landsat images available for each year corresponding to each period of glacier mapping.**

### (2) Establishment of AGEI utilizing the Landsat data cube

In this study, the AGEI is established based on analyses of NDSI, NDWI, and surface temperature. The NDSI is calculated as the difference between the reflectances in the green band and the shortwave infrared 1 (SWIR1) band, divided by the sum of these two reflectances. According to previous studies, the NDSI value typically exceeds 0.4 in areas of snow and ice (Scherler et al., 2018; Huang et al., 2021). Thus, in this study, a threshold value of 0.4 is set to facilitate the extraction of debris-free glaciers from the Landsat images. While the NDSI method effectively excludes most lake pixels at or near the

termini of glaciers, a small number of misidentified pixels persist (Huang et al., 2021). To address this, the AGEI method incorporates the NDWI, a widely used index for extracting water bodies from optical images (McFeeters, 1996; Bevington and Menounos, 2022). The NDWI is calculated as the difference between the reflectances in the green band and the near-infrared (NIR) band, divided by the sum of these two reflectances. Many studies have depicted that the NDWI values of the water pixels ranged from 0.4 to 1 (Du et al., 2016; Zhao et al., 2018; Bevington and Menounos, 2022). Thus, in this study, an

NDWI threshold of 0.4 is adopted to minimize errors associated with the presence of open water in the glacier mapping. Furthermore, glaciers are typically colder than the surrounding seasonal snow or glacial lakes. Therefore, based on prior research and preliminary analysis of the surface temperature of the reference RGI 6.0 glaciers, a threshold for surface



temperature (derived from the thermal band) is set at -1 °C (Shugar et al., 2020). Consequently, the criteria for mapping debris-free glaciers using AGEI involve the use of three indices: an NDSI value greater than 0.4, an NDWI value less than 175  0.4, and a surface temperature lower than -1 °C.

Furthermore, to reduce potential glacier mapping errors stemming from thin clouds or varying terrain conditions, the AGEI method considers the proportion (0-1) of the Landsat data cube that meets the predefined criteria of these three indices. For instance, in a Landsat data cube comprising 100 image scenes, if the NDSI value of a specific pixel exceeds 0.4 in 80 of those scenes, the corresponding proportion for the NDSI is determined as 0.8. The proportion of these three indices is 180  determined iteratively through a process of visual comparison between the mapped glaciers and the optical image mosaic.

**(3) Cleaning and filtering the glacier outlines**

Utilizing the AGEI method previously detailed, the extracted debris-free glaciers are converted into polygons. Subsequently, polygons with an area less than 0.05 km² are excluded, and holes smaller than 0.01 km² are filled (Bevington and Menounos, 2022). The resulting refined debris-free glacier outline is employed for the subsequent analysis of time-series changes.

In this study, the accuracy of the mapped glaciers is validated using reference debris-free glaciers. These reference glaciers are derived by removing the debris-covered portions from the RGI 6.0 dataset, with the debris regions sourced from Scherler et al. (2018). Given that the primary images in RGI 6.0 were predominantly captured between 1999 and 2010 (Pfeffer et al., 2014), the comparative analysis focuses on the glacier mapping results from the periods 2000, 2005, and 2010. To quantitatively evaluate the glacier mapping results, three indices are calculated: Correctness, Completeness, and the F1-score, 190  each based on the reference RGI 6.0 debris-free glaciers. Correctness is computed as the ratio of the correctly mapped glacier area to the total mapped glacier area. Completeness represents the percentage of the correctly mapped glacier area relative to the reference RGI 6.0 debris-free glaciers. F1-score provides a balance between Completeness and Correctness. These indices are defined as follows (Kaushik et al., 2022).

$$Correctness = A_{cg} \, / \, A_{tg} \tag{1}$$

$$Completeness = A_{cg} \, / \, A_{rg} \tag{2}$$

$$F1-score = 2 \times \frac{Correctness \times Completeness}{Correctness + Completeness} \tag{3}$$

where $A_{cg}$ represents the correctly mapped glacier area in this study, $A_{tg}$ is the total mapped glacier area in this study, and $A_{rg}$ is the area of reference RGI 6.0 debris-free glaciers.

**3.2 Downscaling analysis of the ERA5-Land reanalysis data**

To enhance the spatial resolution of air temperature and precipitation data, this study employs the ERA5-Land reanalysis datasets from the European Centre for Medium-Range Weather Forecasts (ECMWF) in the downscaling analysis (Muñoz-Sabater et al., 2021). The ERA5-Land reanalysis datasets, which utilize data assimilation techniques, integrate both ground-based and satellite-derived observations (Essou et al., 2016). ERA5-Land represents the advanced land component of the



fifth-generation European Reanalysis, encompassing data from 1950 to the present, with a spatial resolution of about 9 km
(Muñoz-Sabater et al., 2021). Numerous studies have demonstrated a robust concordance between the ERA5-Land data and
station-based observations (Salerno et al., 2023; Wu et al., 2023; Yilmaz, 2023; Zhou et al., 2023). Although ERA5-Land
data may overestimate precipitation amounts on the Tibetan Plateau, the spatial-temporal patterns of precipitation can be
accurately captured (Salerno et al., 2023; Wu et al., 2023). In this study, the 2-m air temperature and total precipitation from
ERA5-Land are utilized as climate inputs for the downscaling model, and the ancillary variables comprise the MODIS
surface reflectance product, the MODIS land cover product, and the SRTM DEM product.

The downscaling analysis employs the random forest method to establish a relationship between the input ancillary variables
and the meteorological datasets. This method utilizes bootstrap resampling to generate bootstrap samples from the original
data. Each of these samples is subsequently modeled using a decision tree. The collective predictions from these multiple
decision trees are then aggregated. This approach effectively mitigates issues related to outliers in predictions, overfitting,
and missing data in the training samples (Breiman, 2001; Ebrahimy et al., 2021). The downscaling analysis comprises the
following key steps: first, the relationship between the ancillary variables and the ERA5-Land datasets is established at 9 km
resolution using the random forest method; second, this derived relationship is applied to ancillary variables at a finer 1 km
resolution to generate air temperature and precipitation products on a pixel-wise basis; third, the pixel-wise residual error
corresponding to the 9 km resolution is integrated into the spatially coincident downscaled meteorological products. To
examine the impact of climate change on glacier retreat, the downscaled meteorological products are generated for the period
coinciding with glacier mapping, spanning from 1988 to 2022. Annual and seasonal analyses are conducted based on a
hydrologic year that extends from September of one year to August of the next. Accordingly, the autumn season
encompasses September to November, winter spans December to February of the following year, spring covers March to
May, and summer includes June to August.

Further, the trends identified in the downscaled meteorological products are analyzed using the non-parametric Theil-Sen
estimator and Mann-Kendall methods (Some'e et al., 2012; Gocic and Trajkovic, 2013). The Theil-Sen estimator determines
the trend by calculating the slope between pairs of time-series climate data, with the median of these slopes representing the
overall trend in the time-series climate change (Blewitt et al., 2016). The change rate of the time-series datasets is defined as:

$$\beta = Median\left[\frac{X_j - X_i}{j - i}\right] \tag{4}$$

where $\beta$ is the estimated change rate of the time-series data, and $X_i$ and $X_j$ represent sequential data values in the time series
for years $i$ and $j$ ($j>i$), respectively. The Theil-Sen estimator is advantageous as it does not necessitate the time series data to
adhere to the assumptions of serial autocorrelation and normal distribution. Moreover, it is capable of efficiently managing
small outliers and noise from missing values. This robustness has led to its widespread adoption for identifying trend
possibilities in the fields of hydrology and meteorology (Gocic and Trajkovic, 2013; Güçlü, 2018; Prăvălie et al., 2022).





The Mann-Kendall test, a non-parametric method, serves as a complementary tool to the Theil-Sen estimator. It is utilized to test the significance of trends in time-series data, offering a robust approach to confirming the presence and direction of changes within the series (Neeti and Eastman, 2011). The statistic $S$ of the Mann-Kendall test is calculated as follows:

$$S = \sum_{i=1}^{n-1} \sum_{j=i+1}^{n} sgn(X_j - X_i) \tag{5}$$

where $n$ is the number of the time-series data, and $sgn(X_j\text{-}X_i)$ is the sign function, which is formulated as:

$$sgn(X_j - X_i) = \begin{cases} +1 & if \quad X_j - X_i > 0 \\ 0 & if \quad X_j - X_i = 0 \\ -1 & if \quad X_j - X_i < 0 \end{cases} \tag{6}$$

The standard normal test statistic $Z_S$ is estimated as follows:

$$Z_S = \begin{cases} \dfrac{S-1}{\sqrt{Var(S)}} & if \quad S > 0 \\ 0 & if \quad S = 0 \\ \dfrac{S+1}{\sqrt{Var(S)}} & if \quad S < 0 \end{cases} \tag{7}$$

$$Var(S) = \frac{n(n-1)(2n+5) - \sum_{i=1}^{m} t_i(t_i - 1)(2t_i + 5)}{18} \tag{8}$$

where $m$ is the number of the tied groups, and a tied group is a set of sample data having the same value; $t_i$ is the number of ties of extent $i$. In the context of the Mann-Kendall test, a positive $Z_S$ value indicates an increasing trend, whereas a negative $Z_S$ value signifies a decreasing trend. The significance of these trends is determined based on a specific $\alpha$ significance level. In this study, significance levels of $\alpha = 0.05$ and $\alpha = 0.1$ were employed, corresponding to $|Z_S|$ values greater than 1.96 and 1.65, respectively. This trend analysis yields the change rate in the downscaled products along with a measure of their statistical significance. Fig. 3 presents a flowchart that outlines the process for analyzing the impacts of climate change on glacier retreats on the Tibetan Plateau.



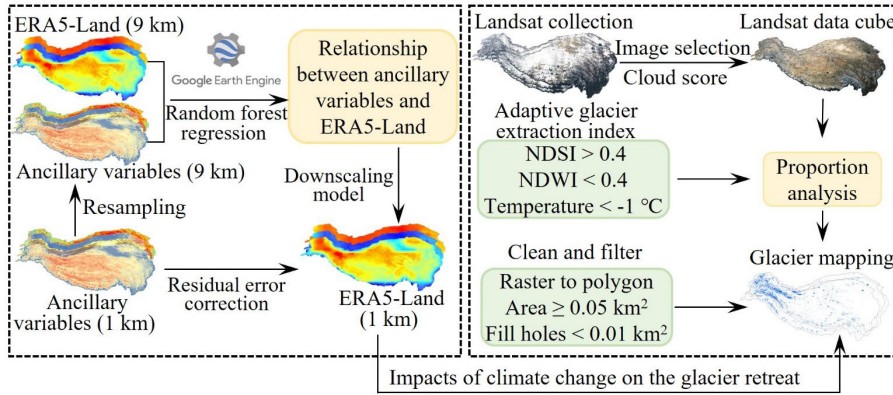

**Figure 3: Flowchart for analyzing the impacts of climate change on the glacier retreat.**

## 4 Results

In this section, the time-series glacier mapping results are first presented; then, the change rates and the significance of the downscaled air temperature and precipitation products are provided; finally, impacts of both annual and seasonal variations in air temperature and precipitation on glacier retreat are quantified.

### 4.1 Time-series glacier mapping results

The glacier area on the Tibetan Plateau, devoid of debris, exhibited a decline from $94.95 \times 10^3$ km² in 1990 to $61.16 \times 10^3$

km² by 2020, corresponding to an average retreat rate of $1.08 \pm 0.28 \times 10^3$ km² per year. Fig. 4 illustrates the spatial changes in glacier extent throughout the observation period. The observed glacier retreat on the Tibetan Plateau is relatively minor in its interior and intensifies towards the periphery. Notably, the most pronounced retreat occurs at the edges, especially in Zones I, II, III, and VIII, situated in the Himalayas and southeastern regions of the plateau. Conversely, glacier advance is predominantly seen in the Karakoram area, a phenomenon referred to as the Karakoram Anomaly. Additionally, a minor

advancement of glaciers is also noted in the transitional area between Zones II and VIII. Fig. 4 reveals marked variations in glacier area change rates across different sub-zones, despite a general trend of shrinkage. During the observation period, the rate of glacier retreat varied from $0.14 \pm 0.07\%$ per year in Zone VI to $0.51 \pm 0.09\%$ per year in Zone III. Notably, a rapid retreat occurred in 2000, particularly in Zones II, V, VI, and VIII. From 2000, there was a slight increase in glacier areas, indicative of glacier advances in 2005 in all zones except for VI and VIII. Since 2010, these sub-zones have exhibited either

a stable or a marginally decreasing glacier area, suggesting a relatively stable glacial state.





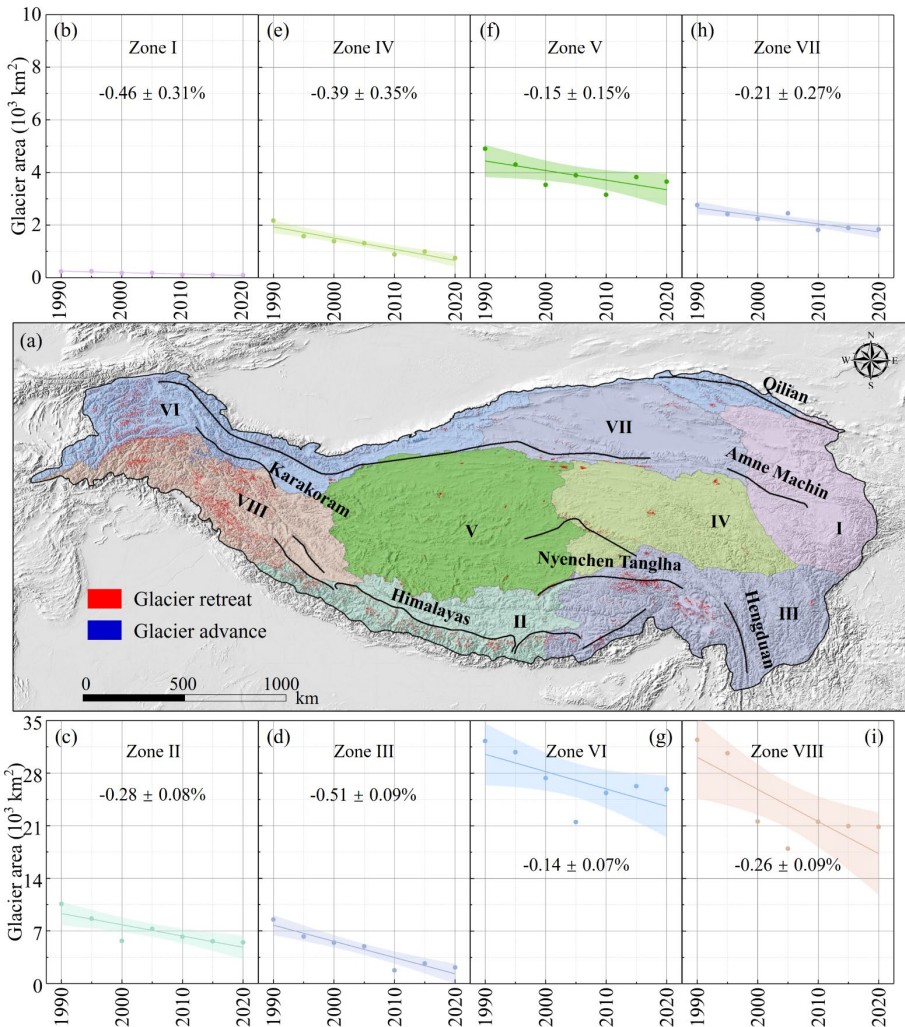

**Figure 4: Spatial changes in glacier extent throughout the observation period in these sub-zones of the Tibetan Plateau: (a) Spatial distributions of glacier-changed areas on the Tibetan Plateau; (b-i) Time-series glacier change rates in Zones I-VIII, respectively (the dot indicates the glacier area in each observation period, and the line indicates the linear fitting of the time-series glacier change restricted with 90% confidence band).**

Fig. 5 presents a comparative analysis of glacier mapping in 2000, 2005, and 2010 against the reference RGI 6.0 debris-free glaciers in these sub-zones. The mapped glaciers in these areas generally align with the debris-free glaciers delineated in RGI 6.0, particularly in Zones I, IV, V, and VII. These sub-zones, located primarily in the interior of the Tibetan Plateau, feature a comparatively smaller glacier area relative to other sub-zones. However, discrepancies between the glaciers mapped in this study and those from RGI 6.0 are observed in Zones VI and VIII. To facilitate a more intuitive comparison of these differences, the average glacier area from the three periods (2000, 2005, and 2010) was utilized for comparison, with the percentage difference between this average and the reference RGI 6.0 detailed in Fig. 5. This analysis reveals the percentage



discrepancy between the average mapped glacier area in this study and that from RGI 6.0 stands at 2.34% in Zone VI and 8.97% in Zone VIII, thereby validating the accuracy of the mapped glaciers in this study. Further, the glaciers mapped in 2000 and 2005 exhibit greater consistency compared to those in 2010, with this trend being particularly pronounced in Zone III. This discrepancy has led to a higher percentage difference in glacier mapping within Zone III. To quantitatively evaluate the glacier mapping results, metrics of Correctness, Completeness, and the F1-score are applied to the mapped glaciers and

the reference RGI 6.0 debris-free glaciers. Table 2 demonstrates that the Correctness of the glaciers mapped during these three periods consistently exceeds 0.7, affirming the accuracy of the mapping results. Furthermore, the majority of glacier outlines in RGI 6.0 were derived from satellite imagery obtained in 1999 or later (Pfeffer et al., 2014). Consequently, the glaciers mapped in 2000 display greater consistency compared to those mapped in 2005 and 2010, which is in agreement with the results depicted in Fig. 5.


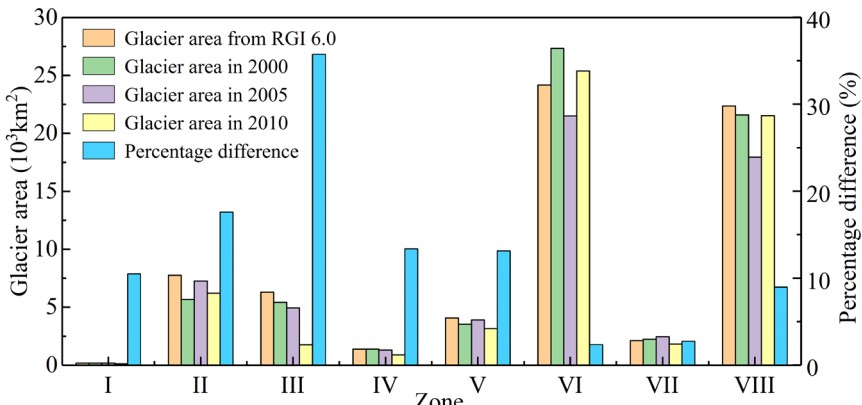

**Figure 5: Comparison results of glacier mapping in 2000, 2005, and 2010 and the reference RGI 6.0 debris-free glaciers.**

**Table 2: Quantitative comparison results of glacier mapping in 2000, 2005, and 2010 and the reference RGI 6.0 debris-free glaciers.**

| Period | Correctness | Completeness | F1-score |
|--------|-------------|--------------|----------|
| 2000 | 0.74 | 0.63 | 0.68 |
| 2005 | 0.71 | 0.55 | 0.62 |
| 2010 | 0.73 | 0.41 | 0.53 |


### 4.2 Downscaled results of the ERA5-Land reanalysis datasets

The ERA5-Land reanalysis datasets, specifically the 2-m air temperature and total precipitation, have been downscaled using the random forest method over the observation period (i.e., from 1988 to 2022). Prior to conducting trend analysis on these downscaled meteorological products, their accuracy is initially evaluated against observational data from meteorological

stations. The locations of these meteorological stations are shown in Fig. 1. Fig. 6 illustrates the comparison between the





downscaled meteorological products and the training samples, validation samples, as well as station observations. The consistency of the downscaled products with these datasets is evident, as reflected by the high values of $R^2$. Therefore, the downscaled meteorological products are deemed suitable for use in subsequent analyses.

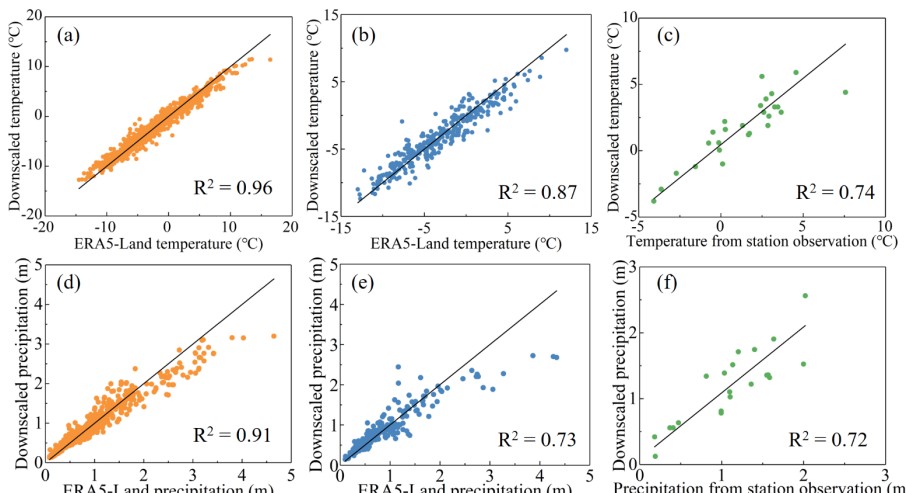

**Figure 6: Comparisons between the downscaled meteorological products and the training samples, validation samples, and station observations: (a-c) Comparative results of the downscaled air temperature with the training samples, validation samples, and station observations, respectively; (d-f) Comparative results of the downscaled precipitation with the training samples, validation samples, and station observations, respectively.**

Fig. 7(a) presents the overall upward trend in 2-m air temperature during the observation period, with the highest increase observed in winter at a rate of 0.26 °C/year, specifically in the eastern Tibetan Plateau (Zone I). However, this warming trend varies across different seasons. Across the Tibetan Plateau, the observed seasonal temperature increase rates are as follows: annual at 0.14°C/decade, autumn at 0.16°C/decade, winter at 0.04°C/decade, spring at 0.008°C/decade, and summer at 0.04°C/decade. Additionally, the rate of increase in 2-m air temperature displays significant spatial heterogeneity. For instance, the areas experiencing the most rapid warming are predominantly located in the eastern and southeastern parts of the Tibetan Plateau, specifically in Zones I, III, and IV, with this trend being particularly pronounced during the fall and winter seasons. Notably, these regions of accelerated warming align with the zones of fastest glacier retreat, as depicted in Fig. 4. Fig. 8(a) illustrates the significance of both annual and seasonal changes in air temperature. It reveals that the regions with significant temperature increases are mainly in the eastern part of the Tibetan Plateau. In contrast, the southwestern and central areas, specifically Zones V and VIII, display a marked decreasing trend in temperature. Correspondingly, as shown in Fig. 4, the glacier retreat rates in these two zones are relatively low. Interestingly, in the transitional area between Zones II and VIII, there is a notable decreasing trend in air temperature during the winter season, which aligns with a minor glacier advance in this region.



In contrast to the 2-m air temperature changes, the total precipitation across the Tibetan Plateau does not exhibit a consistent trend of increase or decrease, as illustrated in Fig. 7(b). The highest increase in precipitation rate, reaching up to 64 mm/year, is observed in the southern part of the plateau, specifically in Zone II. On the other hand, the most significant decrease in precipitation is noted in the southeastern region, identified as Zone III. The average rates of precipitation change vary by season. They were 3.5 mm/decade, 7.0 mm/decade, -5.0 mm/decade, -1.5 mm/decade, and 1.2 mm/decade in annual, fall,

winter, spring, and summer, respectively. Like the 2-m air temperature change, the precipitation variation across the region also demonstrates significant spatial heterogeneity. For instance, a region with a marginal increase in precipitation is predominantly found in the southwest Himalayas and the Karakoram. This increase may contribute to the glacier advance in these areas, a phenomenon also highlighted in Fig. 4. Furthermore, the southeastern Tibetan Plateau, identified as Zone III, exhibits a notable decreasing trend in annual precipitation, which aligns with the observed glacier retreat in this zone.

Additionally, Fig. 8(b) highlights that the annual precipitation markedly increases in the transitional area between Zones II and VIII, whereas it significantly decreases in the southeast Tibetan Plateau.

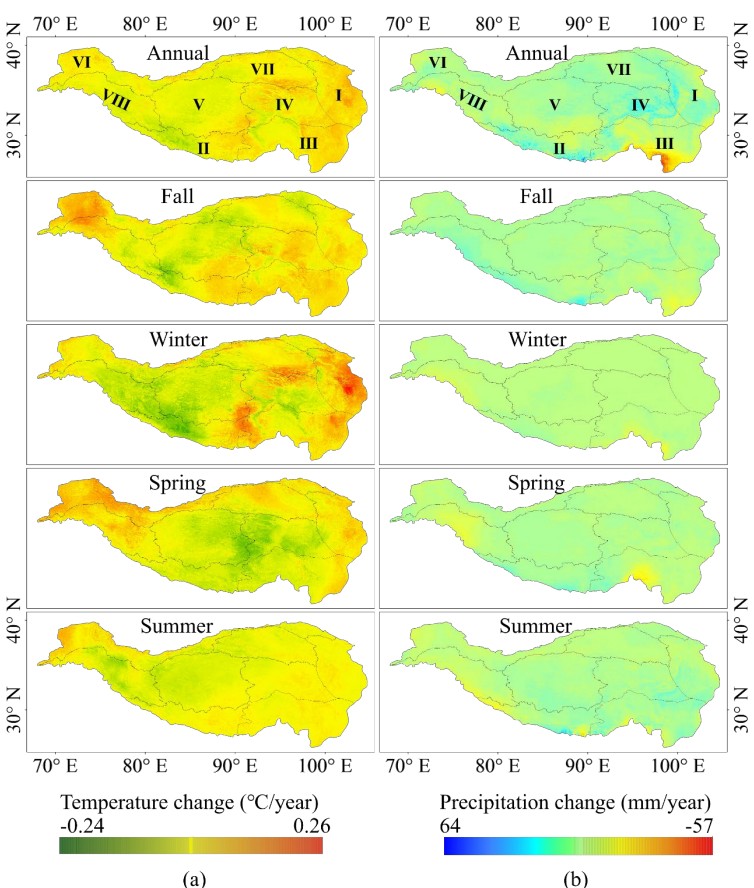

Figure 7: Trend analysis results of the downscaled meteorological datasets: (a) 2-m air temperature; (b) Precipitation.





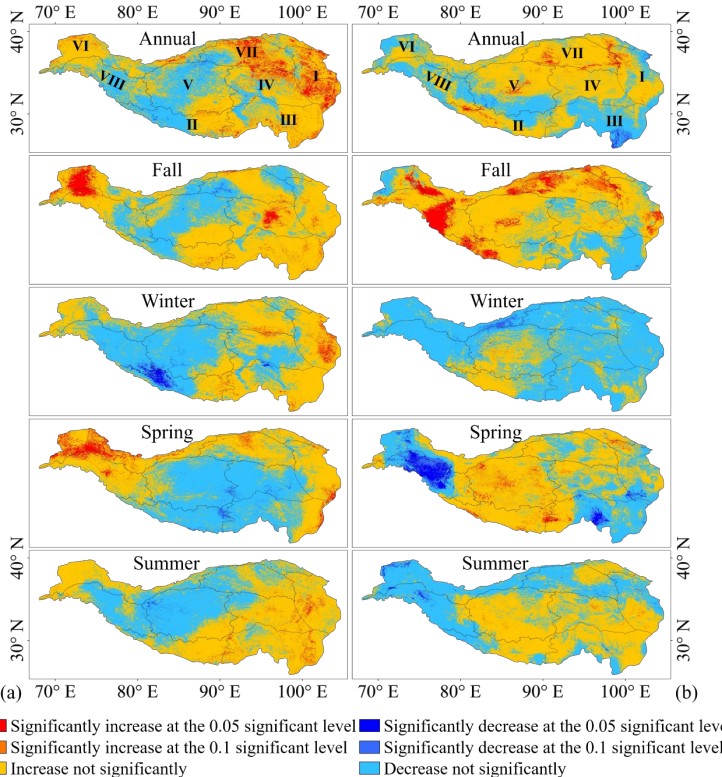

Figure 8: Significance of the trend analysis results: (a) 2-m air temperature; (b) Precipitation.

## 4.3 Impacts of climate change on the glacier retreat

To evaluate the potential effects of the downscaled meteorological factors on glacier retreat, a linear regression model has been employed. This model analyzes the relationships between both annual and seasonal meteorological factors and the changes in glacier area within these sub-zones. The linear regression coefficient is determined by fitting the glacier area changes to the meteorological factors, which include both annual and seasonal air temperature and precipitation, over the period from 1990 to 2020 in each sub-zone. The outcomes of these analyses are presented in Fig. 9. Fig. 9(a) illustrates the impact of downscaled 2-m air temperature on changes in glacier area. Across all the sub-zones, there is a negative correlation between glacier area changes and annual temperature variations, and the most pronounced impacts can be observed in Zone VIII, with a value of $-9.34 \times 10^3$ km$^2$/°C. When considering seasonal variations, the impact of temperature changes on glacier retreat is especially notable in Zones VI and VIII for the winter and summer seasons. The effects of winter air temperatures on glacier change can be $-4.56 \times 10^3$ km$^2$/°C and $-5.16 \times 10^3$ km$^2$/°C in Zones VI and VIII, respectively, and the values for summer air temperature reach $-5.49 \times 10^3$ km$^2$/°C and $-6.46 \times 10^3$ km$^2$/°C. However, the impact of spring temperature on glacier change is less evident except for Zone VIII.





Fig. 9(b) presents the effects of downscaled precipitation on changes in glacier area. It reveals that an increase in annual precipitation can help mitigate glacier retreat except for Zone II. However, this restraining impact of annual precipitation is evident only in Zones III, V, and VIII, with values of 16 km$^2$/mm, 120 km$^2$/mm, and 75 km$^2$/mm, respectively. When considering seasonal variations, the most restraining impact can be observed in Zone VI for the spring season, with a value

of 261 km$^2$/mm. However, the effects of fall precipitation on glacier change in all sub-zones are not apparent. In addition, Fig. 9(b) depicts that the increases in winter and spring precipitations are more influential in reducing glacier retreat compared to those in fall and summer, particularly in Zones III and VIII. Specifically for Zone VIII, the increases in winter and spring precipitations are found to counteract glacier retreat, while increased precipitation in fall and summer might actually accelerate it. This observed phenomenon could be attributed to the different forms of precipitation. In this study, the

total precipitation amount encompasses both rainfall and snowfall, but these two forms have distinct impacts on glacier area changes (Su et al., 2022). Snowfall is beneficial for glacier formation and growth as it increases surface albedo and contributes to mass accumulation on the glaciers. On the other hand, rainfall tends to accelerate glacier retreat due to its influence on the surface energy budget. For instance, rainfall can reduce surface albedo and release latent heat, both of which are factors that contribute to the melting of glaciers.


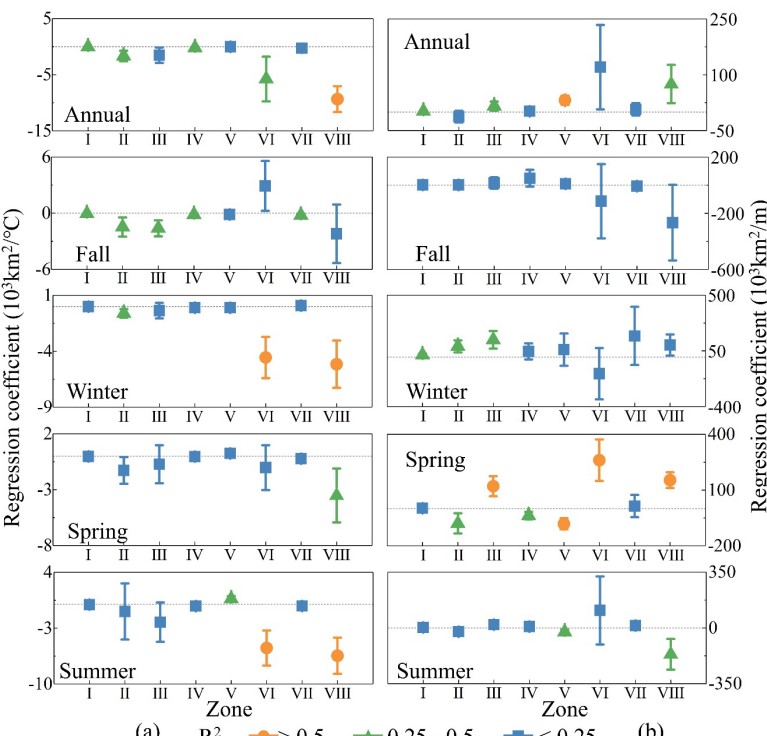

Figure 9: Impacts of the meteorological factors on the glacier area change: (a) 2-m air temperature; (b) Precipitation.



## 5 Discussions

In this section, the advantage of the proposed AGEI method is first clarified; then, the spatial variability in glacier retreat is analyzed; finally, the impact of debris thickness on glacier retreat is demonstrated.

### 5.1 Discussion on the proposed AGEI method

The earlier study utilized the minimum NDSI value at each pixel for glacier mapping, which proved to be particularly effective in minimizing the influence of seasonal snow cover (Huang et al., 2021). However, this approach has limitations in

its robustness, especially when encountering image noises such as small icebergs, brash ice, haze, and issues like the failure of the Scan Line Corrector in Landsat 7 images (Chander et al., 2009; Shugar et al., 2020). Additionally, the cloud score threshold of 60 might have included glacier pixels under thin clouds, potentially leading to NDSI values below 0.4 and affecting glacier delineation accuracy. To mitigate noise and omissions in glacier mapping, the AGEI method is proposed using proportions of the Landsat data cube that meet the predefined criteria of NDSI, NDWI, and surface temperature,

calibrated through visual comparisons with contemporary optical image mosaics. Fig. 10 illustrates a comparison between the glaciers delineated using the AGEI method and the corresponding minimum NDSI values in the period of 2020. The base map is Landsat 8 mosaic images obtained from 2018 to 2022 in the ablation season. It can be seen that the glacier mapping errors caused by glacier lakes can be minimized using the AGEI method compared to the minimum NDSI method, as depicted in Area A; Area B may be affected by thin clouds at a certain period, making the NDSI value at that time less than

0.4, but the glaciers in this area can be better identified by the AGEI method. Furthermore, the AGEI method incorporates surface temperature, a crucial factor when dealing with glacial polygons adjacent to glacial lakes (Shugar et al., 2020). This consideration is vital because glacial lakes, being warmer than the surrounding glaciers and ice, may not be wholly excluded by the NDWI threshold alone.

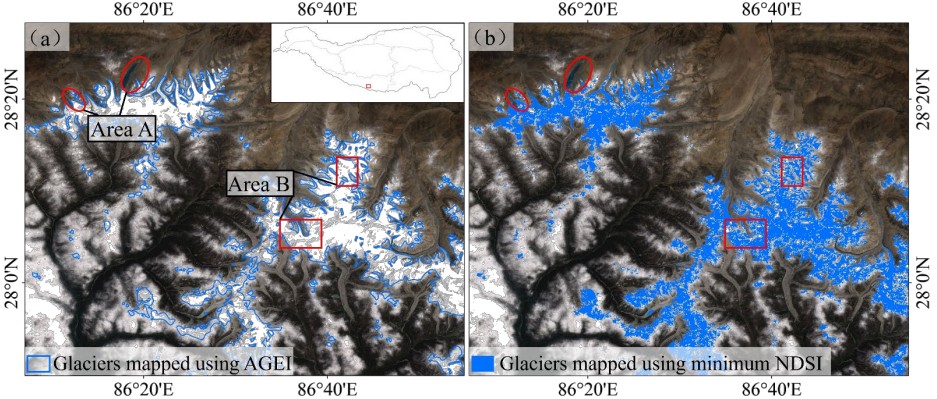

**Figure 10: Comparison between the glaciers delineated using the AGEI method and the corresponding minimum NDSI values: (a) Glaciers mapped using the AGEI method; (b) Glaciers mapped using minimum NDSI values. The base map is the Landsat 8 mosaic images obtained from 2018 to 2022 in the ablation season.**



## 5.2 Discussion of spatial variability in glacier retreat

Fig. 4 highlights the distinct regional variations in glacier area changes, with the most pronounced retreat observed in the Himalayas and the southeastern Tibetan Plateau. Numerous studies corroborate that the retreat in the Himalayan region is marked by the most substantial reductions in glacier length and area, as well as the most negative glacier mass balance (Yao et al., 2012; Latif et al., 2019; Zhang et al., 2021). The rising temperatures and declining precipitation in the southeastern Tibetan Plateau, as depicted in Fig. 8, are likely contributing factors to the glacier retreat observed in this region. Moreover,

a gradient of glacier shrinkage is observed, diminishing progressively from the Himalayas towards the continental interior, indicating a less pronounced retreat in these regions. This area is characterized by the least glacier retreat, minimal area reduction, and a more positive mass balance (see Fig. 4, Yao et al., 2012). This study examines the effects of downscaled air temperature and precipitation on glacier retreat across various sub-zones. However, it does not account for the interactions between air temperature and precipitation. In fact, glaciers are sensitive to the rising air temperature and precipitation that

occurs at temperatures near 0 °C. Such conditions are significant contributors to the loss of snow and ice cover, most notably during the spring season (Kang et al., 2009; Bibi et al., 2018). Therefore, considering the interaction between air temperature and precipitation is essential for a more comprehensive analysis of glacier retreat under climate change. Moreover, atmospheric circulation patterns play a crucial role in dictating the spatial variability of air temperature and precipitation changes, leading to diverse impacts of climate change on glacier retreat across different regions. The sub-zone divisions

illustrated in Fig. 1 incorporate these variations in atmospheric circulation processes. As a result, analyzing glacier changes within these distinct sub-zones offers a more nuanced and reasonable approach to understanding the effects of climate change on glacier retreat.

## 5.3 Discussion on the influence of debris thickness on glacier retreat

This study has shown that the retreat of glaciers, as reflected by changes in glacier area, is primarily due to climate warming.

However, as depicted in Fig. 9, the impacts of climate change on glacier retreat show significant spatial heterogeneity. This variability is primarily attributed to the complex interplay of climatic, geographic, and topographic factors. Among these factors, elements like debris coverage, ice cliffs, and glacial lakes play crucial roles in determining glacier responses to climatic shifts (Johnson and Rupper, 2020; Su et al., 2022; Chen et al., 2023). Notably, the thickness of supraglacial debris is a crucial factor that can significantly influence the rate of glacier retreat in the context of climate change (Pratap et al., 2015).

To explore the potential impacts of supraglacial debris thickness on glacier retreat, the debris thickness datasets are obtained from Rounce et al. (2021). Fig. 11 presents a comparison of the mean and median glacier debris thickness against the glacier retreat rate in various sub-zones. The analysis reveals that areas experiencing rapid glacier retreat, such as Zones I and III, are characterized by lower mean and median values of debris thickness. This observation aligns with research suggesting that thinner debris layers on glaciers lead to increased absorption of solar radiation, thereby accelerating the melting of the glacier

and hastening its retreat (Rounce et al., 2018; Chen et al., 2023). Conversely, in areas with thicker debris layers, such as



Zone V and VI, the thermal insulation provided by the debris predominates, thereby slowing down glacier retreat. The threshold of debris thickness that dictates whether it accelerates or restrains glacier retreat varies across different glaciers. Typically, this threshold ranges from approximately 0.2 m to over 1.0 m and tends to increase as elevation decreases (Chen et al., 2023). In this study, the comparative analysis focuses on different sub-zones, which might yield results that deviate 440 from analyses conducted on individual glaciers. Despite this, considering the response of glacier retreat to debris thickness remains a valuable aspect for further exploration, especially in the context of analyzing single glacier retreat under the influence of climate change.

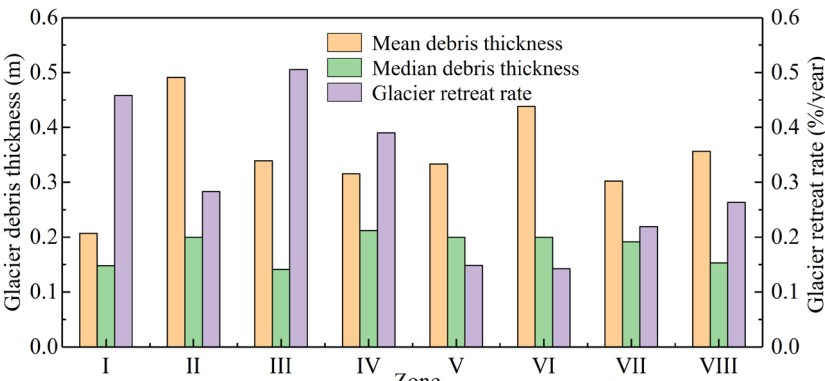

**Figure 11: Comparison of the mean and median glacier debris thickness against the glacier retreat rate in various sub-zones.**

## 6 Conclusions

This study proposed an adaptive glacier extraction index (AGEI) method to map time-series glaciers on the Tibetan Plateau, analyzing glacier retreat with downscaled air temperature and precipitation from 1988 to 2022. The following conclusions are reached based on the results presented.

(1) Utilizing the Google Earth Engine platform and Landsat archives, this study maps the glaciers of the Tibetan Plateau at five-year intervals from 1988 to 2022 using the AGEI method. The total glacier area decreased from $94.59 \times 10^3$ km² to $61.16 \times 10^3$ km², with an average annual retreat rate of $1.08 \pm 0.28 \times 10^3$ km². The rates of glacier retreat varied significantly across the eight sub-zones, ranging from $0.14 \pm 0.07\%$ to $0.51 \pm 0.09\%$ per year over the observation period.

(2) The ERA5-Land reanalysis datasets have been downscaled for this study to examine annual and seasonal trends from 455 1988 to 2022. There is an overall increase in 2-m air temperature during this period, with the maximum rise reaching 0.26 °C/year, particularly in the winter season. The fastest warming areas are predominantly in the eastern and southeastern parts of the Tibetan Plateau. A slight increase in precipitation is mainly observed in the southwest Himalayas and the Karakoram, which may contribute to glacier advancement in these regions.

(3) There is a negative correlation between glacier area changes and annual temperature variations, and the most pronounced 460 impacts are observed in Zone VIII, with a value of $-9.34 \times 10^3$ km²/°C. The impact of winter and summer temperature



changes on glacier retreat is especially notable in Zones VI and VIII. The increase in annual precipitation can help mitigate glacier retreat except for Zone II, and the most restraining impact is observed in Zone VI for the spring season, with a value of 261 km$^2$/mm. In addition, the increases in winter and spring precipitations are more influential in reducing glacier retreat compared to those in fall and summer, particularly in Zones III and VIII.

**Author contribution**

**Fumeng Zhao:** Conceptualization; Data curation; Formal analysis; Methodology; Resources; Software; Writing - original draft; **Wenping Gong:** Conceptualization; Formal analysis; Methodology; Funding acquisition; Writing - original draft & editing; **Silvia Bianchini:** Formal analysis; Resources; Writing - editing; **Zhongkang Yang:** Data curation; Writing - editing.

**Competing interests**

The authors declare that they have no conflict of interest.

**Acknowledgments**

The work was financially supported by the Outstanding Youth Foundation of Hubei Province, China (No. 2022CFA102) and the National Natural Science Foundation of China (No. 42377180).

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
