# Peer review of "Linking Glacier Retreat with Climate Change on the Tibetan Plateau through Satellite Remote Sensing"

_EGUsphere, 2024_

## Author Comment (AC1)

The manuscript presented by Zhao and co-authors put forward a glacier mapping tool using Landsat collections and reanalysis data over the Tibetan Plateau (TP).

The manuscript is generally well written and mostly well structured, however I have made some suggestions to try and improve the flow of the manuscript.

Response: Thank you for your careful review of our manuscript and for your many high-quality comments and suggestions. We sincerely appreciate the comments that have helped sharpen this paper. Specific responses to the review comments are presented immediately after the respective review comments.

My main and minor comments can be found below:

Main:

1. Why is RGI 6.0 being used instead of RGI 7.0 which was published in September 2023? I appreciate if analysis had been carried out before the most recent release, however I think it is important the most up to date data products are used in current day studies as I know there can be large differences in glacier extents between version 6.0 and 7.0 and therefore may impact your results significantly. I will leave this to the decision of the handling editor, but I would suggest using RGI 7.0 to ensure at the time of publication, the results reflect the most current version.

Response: Thank you for your kind comment and suggestion. The RGI 6.0 was the latest glacier data when conducting this work. Thus, the validation of our glacier mapping results is mainly based on the RGI 6.0 glacier data.
The quality of RGI 7.0 data is substantially improved in many regions due to the inclusion of newly updated inventory glacier data (RGI 7.0 Consortium, 2023). Thus, in this revision, the RGI 7.0 data is adopted to validate our glacier mapping results. The validation results indicate that the accuracy of our mapped glaciers in 2000 using the latest RGI 7.0 glacier data is higher than that using the RGI 6.0 glacier data, which further confirms the accuracy of our mapped glaciers. This validation will be added in the potential revision.

2. There is no consideration of the poor 'completeness' value in Table 2, particularly for 2010. The F-1 scores for glacier mapping are moderate, I would not consider them strong metrics in support of the method. It is of course fine to have these scores, however, there needs to be consideration of why the scores are this low and how this may impact the results in the discussion.

Response: Thank you for your comment. The target year of RGI glacier data is 2000; thus, the completeness of the mapped glaciers in our study was particularly poor in 2010. This consideration will be added in the potential revision.
Further, 35% of all RGI 6.0 outlines were dated to five or more years away from the target year 2000, while this number is down to 23% in RGI 7.0 (RGI 7.0 Consortium, 2023). In addition, the accuracy of our mapped glaciers in 2000 using the latest RGI 7.0 glacier data is higher than that using the RGI 6.0 glacier data, which further confirms the accuracy of our mapped glaciers. In the potential revision, the comparison results between the mapped glaciers in this study and the RGI 6.0

and RGI 7.0 glacier data will also be added.

3. Each section of the paper (i.e. results, discussion) does not need to have an introductory section of how it is structured and I would suggest deleting them and going straight into the text of the section.

Response: Thank you for your comment and suggestion. These introduction sections will be deleted in the potential revision.

Minor:

4. L17: Would just note what time period this 'slight increase' was

Response: Thank you for your comment. Most glacier areas experienced a decrease from 1990 to 2000, followed by a slight increase from 2000 to 2010. The "slight increase" will be clarified in the potential revision.

5. L19: Would consider stating geographically where these 'zones' are as out of context as they don't mean very much as no lead in from the abstract (i.e. NE of TP)

Response: Thank you for your kind suggestion. The southwestern Tibetan Plateau will replace Zone VIII, and the northwest Tibetan Plateau will replace Zone VI in the potential revision.

6. L35: I get what you mean, but I would rephrase the opening on this paragraph saying 'Glacier change can be measured using variations in...' to be clearer

Response: Thank you for your kind suggestion. This sentence will be revised as follows: Glacier change can be measured using a spectrum of metrics, including variations in area, thickness, volume, surface mass balance, and equilibrium line altitude.

7. L42: Would put the reference for each method after it was referred to i.e., spectral analysis (ref), object-based (ref), etc.

Response: Thank you for your comment. This sentence will be modified as follows: Glaciers are commonly mapped using a variety of techniques, including spectral analysis of optical satellite images (Bolch et al., 2010), object segmentation-based methods (Robson et al., 2015), and supervised machine-learning algorithms (Khan et al., 2020).

8. L46: needs a reference for debris-free glaciers before full stop

Response: Thank you for your kind comment. The reference for debris-free glaciers will be added as follows: Robust and efficient pixel-based multispectral analysis has been particularly effective in accurately delineating debris-free glaciers (Huang et al., 2021).

9. L51: are 'quality' and 'resolution' not the same thing? Also, what type of resolution, spatial or temporal? If both I would suggest stating the limitation of spatio-temporal resolution over the TP.

Response: Thank you for your comment and suggestion. There are some differences between quality and resolution; specifically, the resolution is one type of quality. However, in this study, they both indicate the spatial and temporal resolutions of the glacier mapping on the Tibetan Plateau. However, glacier mapping on the Tibetan Plateau with high spatial-temporal resolutions is limited due to the large amounts of satellite images and the massive computing. To avoid this confusion, this sentence will be revised as follows: Furthermore, limited by the large amounts of satellite images and massive computing, the comprehensive depiction of glacier retreat across the entire Tibetan Plateau, especially at finer temporal and spatial resolutions, remains inadequately characterized.

10. L58: Climate change seems a bit broad here, is it the increasing air temperature? Would clarify

Response: Thank you for your comment and suggestion. The dominant driver of the glacier retreat is the increasing temperature. To avoid this confusion, this sentence will be revised as follows: Climate change, especially the increasing temperature, is recognized to be the dominant driver of the glacier mass balance and the associated area and volume changes.

11. L77: If you state 'numerous studies' I would suggest citing a handful of them as examples

Response: Thank you for your comment. More references will be added in the potential revisions, and the revised sentence is as follows: Despite numerous studies examining glacier variations on the Tibetan Plateau in recent decades (Yao et al., 2012; Neckel et al., 2014; Ye et al., 2017; Bibi et al., 2018; Sun et al., 2018; Latif et al., 2019; Zhang et al., 2021; Xiao et al., 2023), the specific impacts of climate change on glacier retreat have not been thoroughly investigated at a finer resolution.

12. L82-85: not convinced this is required for such a specific structure, think you're fine with just the aims of the paper being highlighted

Response: Thank you for your kind comment and suggestion. These sentences are intended to understand the structure of this article. To avoid this confusion, these sentences will be deleted in the potential revision.

13. L86: Just call it Study Area

Response: Thank you for your comment. This subheading will be revised as "Study Area".

14. L117: Would call it data and methods instead of materials

Response: Thank you for your comment. This subheading will be revised as "Data and Methods".

15. L118: This text is not needed - would go straight to 3.1

Response: Thank you for your comment. In the potential revision, this text will be deleted.

16. L124: I would just be careful saying images via GEE catalog are 'open-access' - while they are for individuals, there is commercial cost to access the platform

Response: Thank you for your comment and suggestion. In the potential revision, the "open-access" will be deleted to avoid this confusion, and the revised sentence is as follows: The Landsat data is used via the Google Earth Engine platform, which is attributed to their prolonged data availability period and comparatively high spatial resolution.

17. L124: Would merge these sentences, suggest at start of sentence saying Landsat data is used via GEE

Response: Thank you for your comment and suggestion. In the potential revision, this sentence will be revised as follows: The Landsat data is used via the Google Earth Engine platform, which is attributed to their prolonged data availability period and comparatively high spatial resolution.

18. L158: Figure 2 caption - Is this the total number of Landsat images or the total number used in the study with less than 60% cloud as defined by your study? Would clarify

Response: Thank you for your comment. The number of Landsat images is the total number used in the study with less than 60% cloud. In the potential revision, this sentence will be revised as follows: The number of Landsat images with less than 60% cloud available for each year corresponding to each period of glacier mapping is depicted in Fig. 2. In addition, the caption of Fig. 2 will be revised as follows: Number of Landsat images with less than 60% cloud for each year corresponding to each period of glacier mapping.

19. L162-164: basically, the same as previous, would suggest merging and stating previous studies chose 0.4 as a threshold and therefore it was chosen here

Response: Thank you for your kind comment and suggestion. In the potential revision, this sentence will be revised as follows: Previous studies chose 0.4 as a threshold to extract snow and ice (Scherler et al., 2018; Huang et al., 2021); thus, in this study, a threshold value of 0.4 is set to facilitate the extraction of debris-free glaciers from the Landsat images.

20. L167-169: same applies about thresholding values NDWI, just merge them and say you chose 0.4

Response: Thank you for your kind comment and suggestion. In the potential revision, this sentence will be revised as follows: Many studies have depicted that the NDWI values of the water pixels ranged from 0.4 to 1 (Du et al., 2016; Zhao et al., 2018; Bevington and Menounos, 2022); thus, in this study, an NDWI threshold of 0.4 is adopted to minimize errors associated with the presence of open water in the glacier mapping.

21. L171: 'Therefore' does not fit here as does not follow the previous sentence

Response: Thank you for your comment. In the potential revision, this sentence will be revised as follows: Based on prior research and preliminary analysis of the surface temperature of the reference RGI 7.0 glaciers, a threshold for surface temperature (derived from the thermal band) is set at -1 °C (Shugar et al., 2020).

22. L173-174: the final paragraph is just repetition of values you've defined - suggest deleting

Response: Thank you for your kind suggestion. To avoid this repetition, this sentence will be deleted in the potential revision.

23. L183: What holes? What do you mean 'filled'? Interpolated? Would clarify

Response: Thank you for your comment. These holes may be a part of the mapped individual glaciers. Limited by the spatial resolution of the used Landsat images and the processing errors, there may be some holes in the mapped individual glacier. In this study, these holes smaller than 0.01 km² are filled according to previous studies (Bevington and Menounos, 2022); as such, a more complete glacier mapping can be obtained. To avoid this confusion, this sentence will be revised as follows: Limited by the spatial resolution of the used Landsat images, there may be some errors and holes in the mapped individual glacier. In this study, polygons with an area less than 0.05 km² are excluded, and holes smaller than 0.01 km² are filled according to previous studies (Bevington and Menounos, 2022).

24. L185: sentence does not make sense - what do you mean 'validated using reference debris-free glaciers'?

Response: Thank you for your comment. The mapped glaciers in this study are debris-free glaciers; thus, the accuracy of the mapped debris-free glaciers is validated using reference RGI 7.0 debris-free glaciers. These reference glaciers are derived by removing the debris-covered portions from the RGI 7.0 dataset, with the debris regions sourced from Scherler et al. (2018). In the potential revision, this sentence will be revised as follows: The mapped glaciers in this study are debris-free glaciers; thus, the accuracy of the mapped debris-free glaciers is validated using reference RGI 7.0 debris-free glaciers. These reference glaciers are derived by removing the debris-covered portions from the RGI 7.0 dataset, with the debris regions sourced from Scherler et al. (2018).

25. L253: Can Fig 3 be made bigger?

Response: Thank you for your comment and suggestion. In the potential revision, Fig. 3 will be made bigger.

26. L255-258: Delete introductory

Response: Thank you for your comment. In the potential revision, this paragraph will be deleted.

27. L275: Figure 4. Would suggest having the map as a) at the top of the figure then having panels b-i stacked. The lettering order seems a little confusing

Response: Thank you for your comment and suggestion. In the potential revision, the order will be re-sorted alphabetically.

28. L287: What do you mean the glaciers mapped between 2000 and 2005 'exhibit greater consistency'?

Response: Thank you for your comment. Fig. 5 depicts the comparison results of glacier mapping in 2000, 2005, and 2010 and the reference RGI 6.0 debris-free glaciers. It describes that the glaciers mapped in 2000 and 2005 exhibit greater consistency compared to those in 2010, with this trend being particularly pronounced in Zone III. In the potential revision, the latest RGI 7.0 will be used as the reference glaciers, and the comparisons between the mapped glaciers in this study and the RGI 7.0 glaciers will be depicted.

29. L300: Table 2 metric scores - see main comment 2

Response: Thank you for your comment. The target year of RGI glacier data is 2000; thus, the completeness of the mapped glaciers in our study was particularly poor in 2010. This consideration will be added in the potential revision. Further, 35% of all RGI 6.0 outlines were dated to five or more years away from the target year 2000, while this number is down to 23% in RGI 7.0 (RGI 7.0 Consortium, 2023). Thus, the accuracy of our mapped glaciers in 2000 using the latest RGI 7.0 glacier data is higher than that using the RGI 6.0 glacier data, which further confirms the accuracy of our mapped glaciers. In the potential revision, the comparison results between the mapped glaciers in this study and the RGI 6.0 and RGI 7.0 glacier data will also be added.

30. L400: I think Fig.10 and above text is more results than discussion. Would suggest moving to results. Do you have any numbers for the total area difference between the two methods? Would add to the argument of the AGEI method

Response: Thank you for your comment and suggestion. In the potential revision, the comparison results of the mapped glacier using the AGEI method and the minimum NDSI method will be moved to the results section, and the quantitative comparison between these two methods will also be added in the results section.

31. L405: Would refrain from starting discussion sentences 'Fig X', if wanting to directly refer to the figure, place in brackets at the end of the sentence

Response: Thank you for your kind suggestion. In the potential revision, this sentence will be revised as follows: The distinct regional variations in glacier area changes, with the most pronounced retreat observed in the Himalayas and the southeastern Tibetan Plateau (see Fig. 4).

---

## Author Comment (AC2)

GENERAL COMMENTS

This manuscript investigates how changes in debris-free glacier extent in the Tibetan Plateau between ~1990-2020 are related to changes in climate (temperature, precipitation) over the same period. The authors: i) outline a novel technique (the AGEI) for mapping glacier extent, and use it to quantify changes in glacier area at 5-year intervals for 8 subzones of the Tibetan Plateau; ii) downscale ERA5-Land reanalysis data (specifically 2 m temperature and total precipitation) from 9 $km^2$ to 1 $km^2$ using ancillary variables from MODIS and the SRTM DEM) in a random forest regression, and; iii) use linear regressions to investigate the relationships between changes in glacier extent, temperature and precipitation at annual and seasonal scales. The authors conclude that total (debris-free) glacier area in the region has reduced over the survey duration, that temperatures have typically risen, that precipitation has increased in some subzones and reduced in others, and that in some subzones (particularly those in the Karakorum) reductions in glacier area correlated strongly with increases in temperature, but that the severity of glacier area loss may have been mitigated by corresponding increases in precipitation.

The manuscript presents a significant body of work, including novel mapping methodologies, extensive glacier and climate datasets, and insights into regional glacier and climate change in the Tibetan Plateau, and should therefore be of interest to others mapping glacier change generally, and particularly researchers investigating glacier change in central Asia. The addition of a paragraph discussing the implications of the glacier - climate relationships identified in this work with regard to existing climate projections for the region could help add impact. The manuscript is largely well-written, structured and presented, and could be further improved with some edits to make it more succinct (see comments for specific guidance).

My main comments relate to the methods adopted, several of which require additional clarification and/or detail to by fully comprehensible and transparent. I think this is of particular importance given the novelty of the mapping method used and the borderline nature of some of the glacier - climate relationships presented (e.g. see confidence intervals in Fig. 9). A caveat to my comments is that I am not an expert in Himalayan/Tibetan glaciation, or well-versed in random forest or time-series analyses, and have thus taken the methods adopted in section 3.2 (specifically lines 211-250) in good faith.

Response: Thank you for your careful review of our manuscript and for your many high-quality comments and suggestions. We sincerely appreciate the comments and suggestions that will help sharpen this paper. In the potential revision, the sixth phase of the coupled model intercomparison project (CMIP6) will be adopted to assess future climate change, and three scenarios (SSP1-2.6, SSP2-4.5, and SSP5-8.5) will be used for future climate projections from 2030 to 2100, as such, the future glacier area change will be discussed under different climate change scenarios.

Further, the linear regression method will be clarified in the potential revision, and the detailed results of the regression (i.e., Fig. 9) will be presented in a supplement table. Then, the impact of the annual and seasonal temperature and precipitation change on the glacier area change in different sub-zones of the Tibetan Plateau will be re-discussed. Specific responses to the review comments are given immediately after the respective review comments.

SPECIFIC COMMENTS

1. The accuracy and consistency of the novel AGEI mapping technique is integral to the results of this analysis. Consequently, a more detailed evaluation of the technique would be welcome, both to aid replication and assess its effectiveness. In particular, Fig. 10 does not have sufficient resolution to assess the accuracy of the AGEI mapping. Instead, a figure with multiple high-resolution panels (including some features that present challenges to accurate glacier mapping, such as lakes, shadow, cloud etc.), would be useful to illustrate the accuracy of the AGEI glacier outlines, particularly if the reference RGI 6.0 glacier outlines can be overlaid on them. Given its importance to the results, I think such a figure would be better situated and discussed in Section 3.1 or Supplementary Information, rather than waiting until the discussion.

In addition, I struggled to fully understand the mapping evaluation indices presented on lines 189-198, particularly the Correctness (differentiating between Acg and Atg) and F1 scores (e.g. what does a high F1 score indicate?). If possible, illustrations of high and low scoring glaciers/regions/test sites (e.g. in Supplementary Information) could aid interpretation of the indices and the demonstrate the accuracy of the AGEI technique.

Response: Thank you for your kind comment and suggestion. In the potential revision, a detailed evaluation of the AGEI method will be added in Section 3.1. Particularly, the section of discussion on the proposed AGEI method will be moved to the results part, and a figure with multiple high-resolution panels (including some features that present challenges to accurate glacier mapping, such as lakes, shadows, cloud, etc.) will be added to illustrate the accuracy of the AGEI method.

Further, the three mapping indices (Correctness, Completeness, and F1-score) will be described in Section 3.1. For example, Correctness is computed as the ratio of the correctly mapped glacier area to the total mapped glacier area in this study, and the F1 score provides a balance between Completeness and Correctness, and the high F1 score indicates the high accuracy of our mapped glaciers when using the RGI glacier as the reference glacier. For illustration purposes, the high and low-scoring glacier test sites will be added in the supplement information.

2. An additional section should be added to the end of the methodology to describe the linear       regressions       used       to       investigate       the       relationships       between

temperature/precipitation and glacier change (the linear model is mentioned at the beginning of Section 4.3, but not described fully).

For transparency it would be useful to write out the model equation(s) used in the methods. It is unclear whether numerous independent models were generated (e.g. one for each for each climate variable, zone and season, totalling ~80 independent regressions), or if these were all included in one multiple regression model (and, if so, how multicollinearity between the seasonal climate was addressed).

The results of the regressions should also be presented in a table that includes the coefficients, confidence intervals, standard error, t-values and p-values, and a summary of model diagnostics (R2/Adjusted R2, RSE etc.). Although the coefficients and confidence intervals are presented in Fig. 9 their numeric values are difficult to determine, hence a table of the regression outputs would be useful to support this figure, aid interpretation of the data, and provide more robust conclusions.

Response: Thank you for your kind comment and suggestion. In the potential revision, the linear regression method adopted will be detailed at the end of Section 3.

In this study, the impacts of annual and seasonal temperature and precipitation on the glacier area change are included in one regression model for simplified display. Though studies showed that there is a positive scaling between extreme precipitation and surface air temperature (Yong et al., 2021), the relationship between precipitation and temperature is complex due to the unique geographical and climatic features, and the correlation between the temperature and precipitation is not significant, particularly on an annual or seasonal basis (Duan and Xiao, 2015; Wu et al., 2015). In addition, several studies have proved the effectiveness of the separate impacts of temperature and precipitation on the glacier retreat (Li et al., 2019; Bevington and Menounos, 2022). Thus, the interactions between air temperature and precipitation were not accounted for in this analysis. In the future study, the interaction between air temperature and precipitation will be considered in analyzing the impacts of climate change on the glacier retreat.

In the potential revision, the table presenting the regression results will be added in the Supplement information. In the supplementary table, the coefficients, confidence intervals, standard error, t-values, p-values, and a summary of model diagnostics ($R^2$, Adjusted $R^2$, and RSE) will be displayed.

Bevington, A. R., and Menounos, B.: Accelerated change in the glaciated environments of western Canada revealed through trend analysis of optical satellite imagery. Remote Sens Environ., 270, 112862, https://doi.org/10.1016/j.rse.2021.112862, 2022.
Duan, A., and Xiao, Z.: Does the climate warming hiatus exist over the Tibetan Plateau? Sci Rep., 5(1), 13711, https://doi.org/10.1038/srep13711, 2015.
Li, Y. J., Ding, Y. J., Shangguan, D. H., and Wang, R. J.: Regional differences in global

glacier retreat from 1980 to 2015. Adv. Clim. Change Res., 10(4), 203-213, https://doi.org/10.1016/j.accre.2020.03.003, 2019.

Wu, G., Duan, A., Liu, Y., Mao, J., Ren, R., Bao, Q., and Hu, W.: Tibetan Plateau climate dynamics: recent research progress and outlook. Natl Sci Rev., 2(1), 100-116, https://doi.org/10.1093/nsr/nwu045, 2015.

Yong, Z., Xiong, J., Wang, Z., Cheng, W., Yang, J., and Pang, Q.: Relationship of extreme precipitation, surface air temperature, and dew point temperature across the Tibetan Plateau. Clim Change., 165, 41, https://doi.org/10.1007/s10584-021-03076-2, 2021.

3. A major caveat of this work is that the analysis presented only considers debris-free glaciers. Greater context regarding the prevalence and distribution of debris-covered glaciers in the region would therefore be useful (in the introduction, or sections 3.1 or 5.3) to understand the potential implications of this methodological choice on the results and their interpretation (e.g. are debris-free glaciers typically representative of glaciers across the region; could there be some sub-zones where the exclusion of debris-covered glaciers from the analysis significantly reduces the sample size and therefore the representativeness of the results?).

Response: Thank you for your kind comment. Using the glacier boundaries of RGI, it was estimated that 10% of glaciers are covered by debris on the Tibetan Plateau and its surrounding areas (Scherler et al. 2018), and the debris-covered glaciers on the Tibetan Plateau are mainly distributed in the Himalayas (Ojha et al., 2017; Ji et al., 2022). The debris layer makes the melting process of glaciers complicated, and the debris-covered glaciers have contrasting melting mechanisms and climate response patterns if compared with debris-free glaciers (Chen et al., 2023; He et al., 2023). Thus, the impacts of climate change on the debris-free and debris-covered glaciers retreats are usually analyzed separately (Hu et al., 2022).

However, it is difficult to recognize debris automatically from satellite images. Only when glacier boundaries are given can the discrimination of debris from snow and ice become operational (Huang et al., 2022). Considering that there is no accurate method to extract debris change, thus, in this study, glacier area change only refers to changes in debris-free glaciers, and the exclusion of debris-covered glaciers from the analysis will not reduce the sample size. To avoid this confusion, the distribution and discussion of debris-covered glaciers will be added in the potential revision.

Chen, F., Wang, J., Li, B., Yang, A., and Zhang, M.: Spatial variability in melting on Himalayan debris-covered glaciers from 2000 to 2013. Remote Sens Environ., 291, 113560, https://doi.org/10.1016/j.rse.2023.113560, 2023.

He, Z., Yang, W., Wang, Y., Zhao, C., Ren, S., and Li, C.: Dynamic changes of a thick debris-covered glacier in the southeastern Tibetan Plateau. Remote Sens., 15(2), 357, https://doi.org/10.3390/rs15020357, 2023.

Hu, M., Zhou, G., Lv, X., Zhou, L., Wang, X., He, X., and Tian, Z.: Warming Has Accelerated the Melting of Glaciers on the Tibetan Plateau, but the Debris-Covered Glaciers Are Rapidly Expanding. Remote Sens., 15(1), 132, https://doi.org/10.3390/rs15010132, 2022.

Huang, L., Li, Z., Zhou, J. M., and Zhang, P.: An automatic method for clean glacier and nonseasonal snow area change estimation in High Mountain Asia from 1990 to 2018. Remote Sens Environ., 258, 112376, https://doi.org/10.1016/j.rse.2021.112376, 2021.

Ji, Q., Yang, T. B., Li, M. Q., Dong, J., Qin, Y., and Liu, R.: Variations in glacier coverage in the Himalayas based on optical satellite data over the past 25 years. Catena, 214, 106240, https://doi.org/10.1016/j.catena.2022.106240, 2022.

Ojha, S., Fujita, K., Sakai, A., Nagai, H., and Lamsal, D.: Topographic controls on the debris-cover extent of glaciers in the Eastern Himalayas: Regional analysis using a novel high-resolution glacier inventory. Quat. Int., 455, 82-92, https://doi.org/10.1016/j.quaint.2017.08.007, 2017.

Scherler, D., Wulf, H., and Gorelick, N.: Global assessment of supraglacial debris-cover extents. Geophys Res Lett., 45(21), 11-798, https://doi.org/10.1029/2018GL080158, 2018.

4. Given the relationships identified between changes in glacier area, temperature and precipitation from ~1990-2020, I think a paragraph could be added to the discussion to explore the implications of these results in respect of existing climate projections for the region. For example, how is regional climate likely to change by the end of the century and what could be the implications for the glaciers in the different sub-zones given the trends identified between 1990-2020? A short paragraph addressing this could help to add impact to the manuscript, and potentially identify future research priorities.

Response: Thank you for your kind comment and suggestion. In the potential revision, the sixth phase of the coupled model intercomparison project (CMIP6) will be adopted to assess future climate change, and three scenarios (SSP1-2.6, SSP2-4.5, and SSP5-8.5) will be used for future climate projections from 2030 to 2100. As such, the average temperature and precipitation change in the eight sub-zones of the Tibetan Plateau will be displayed by the end of the century, and the future glacier area change under three climate change scenarios will be discussed.

5. Throughout the manuscript the authors refer to 'glacier retreat' which (to me at least) implies a measure of distance retreated by a glacier terminus, particularly the terminus of a valley or outlet glacier. Given that the work presented is concerned with changes in glacier area (not just recession at termini) 'glacier extent' may be a more intuitive term to use.

Response: Thank you for your kind comment and suggestion. This study mainly

analyzed the glacier area change on the Tibetan Plateau, which is also the change in glacier extent. However, Glacier retreat can be measured using reductions in area, thickness, volume, surface mass balance, and increase of equilibrium line altitude (Sugiyama et al., 2013; Su et al., 2022). The retreat of glaciers located on the Tibetan Plateau was characterized by severe area shrinkage and mass loss (Li et al., 2019). In addition, many studies have depicted glacier retreat using glacier area change (Dyurgerov et al., 2009; Li et al., 2019; Sommer et al., 2020). To avoid this confusion, the glacier area change will be described as the change of glacier extent, and the glacier retreat will be used in some discussion sections.

Dyurgerov, M., Meier, M. F., and Bahr, D. B.: A new index of glacier area change: a tool for glacier monitoring. J Glaciol., 55(192), 710-716, https://doi.org/10.3189/002214309789471030, 2009.

Li, Y. J., Ding, Y. J., Shangguan, D. H., and Wang, R. J.: Regional differences in global glacier retreat from 1980 to 2015. Adv. Clim. Change Res., 10(4), 203-213, https://doi.org/10.1016/j.accre.2020.03.003, 2019.

Sommer, C., Malz, P., Seehaus, T. C., Lippl, S., Zemp, M., and Braun, M. H.: Rapid glacier retreat and downwasting throughout the European Alps in the early 21st century. Nat Commun, 11(1), 3209, https://doi.org/10.1038/s41467-020-16818-0, 2020.

Su, B., Xiao, C., Chen, D., Huang, Y., Che, Y., and Zhao, H.: Glacier change in China over past decades: Spatiotemporal patterns and influencing factors. Earth-Sci. Rev., 226, 103926, https://doi.org/10.1016/j.earscirev.2022.103926, 2022.

Sugiyama, S., Fukui, K., Fujita, K., Tone, K., and Yamaguchi, S.: Changes in ice thickness and flow velocity of Yala Glacier, Langtang Himal, Nepal, from 1982 to 2009. Ann. Glaciol., 54(64), 157-162, https://doi.org/10.3189/2013AoG64A111, 2013.

6. The manuscript could be made more succinct by removing several sections of text outlining the structure (which is self-evident), particularly lines 82-85, 118-119, 255-257, and 380-381.

Response: Thank you for your kind comment and suggestion. These introduction sections will be deleted in the potential revision.

7. Figures 4 & 5: Both figures show an increase in glacier area in zones II, V and VII between 2000 and 2005. This seems unusual given the overall regional trend, so I'm interested to know whether this is an artefact of the mapping methodology, or whether this brief increase in glacier extent in these regions has been documented in any other studies/literature? The same question also applies to zones VI and VIII, which appear to show notable increases in glacier area between 2005-2010.

Response: Thank you for your comment. Such glacier area increases are also documented in the references. For example, some studies show that there is a slight

increase in glacier area in East Kunlun, Inner Tibet, and Central Himalayas from 2000 to 2005 (Huang et al., 2021); these areas correspond to Zones II and V. In addition, some studies show that the glacier area in the Karakoram (i.e., Zones VI and VIII) increased from 2006 to 2010 (Yao et al., 2012). To avoid this confusion, the glacier area increase in these zones will be detailed in the potential revision.

Huang, L., Li, Z., Zhou, J. M., and Zhang, P.: An automatic method for clean glacier and nonseasonal snow area change estimation in High Mountain Asia from 1990 to 2018. Remote Sens. Environ., 258, 112376, https://doi.org/10.1016/j.rse.2021.112376, 2021.
Yao, T., Thompson, L., Yang, W., Yu, W., Gao, Y., Guo, X., and Joswiak, D.: Different glacier status with atmospheric circulations in Tibetan Plateau and surroundings. Nature Clim Change, 2(9), 663-667, https://doi.org/10.1038/nclimate1580, 2012.

TECHNICAL COMMENTS
10. Abstract: Here, references to the zones used in the study would be better substituted with the names of the corresponding regions (e.g. Karakorum, southern Himalaya etc.) because the zones are only intelligible to those who have already read Section 2.

Response: Thank you for your kind comment and suggestion. The southern Himalayas will replace Zone VIII, and the Karakoram will replace Zone VI in the potential revision.

11. Lines 209-210: Is there a precedent or justification for the use of the 3 ancillary MODIS and SRTM DEM variables to downscale the ERA5-Land data?

Response: Thank you for your comment. Yes, the ERA5-Land is commonly downscaled based on ancillary factors, these ancillary factors include MODIS surface reflectances, NDVI, and DEM data (Kusch and Davy, 2022; Karaman and Akyürek, 2023; Wang et al., 2023). In the potential revision, more references will be cited to support the downscaling analysis.

Karaman, Ç. H., and Akyürek, Z.: Evaluation of near-surface air temperature reanalysis datasets and downscaling with machine learning based Random Forest method for complex terrain of Turkey. Adv. Space Res., 71(12), 5256-5281, https://doi.org/10.1016/j.asr.2023.02.006, 2023.
Kusch, E., and Davy, R.: KrigR-a tool for downloading and statistically downscaling climate reanalysis data. Environ. Res. Lett., 17(2), 024005, https://doi.org/10.1088/1748-9326/ac48b3, 2022.
Wang, N., Tian, J., Su, S., and Tian, Q.: A Downscaling Method Based on MODIS Product for Hourly ERA5 Reanalysis of Land Surface Temperature. Remote Sens., 15(18), 4441, https://doi.org/10.3390/rs15184441, 2023.

12. Line 247: A significance level of 0.1 seems unusually low, and unnecessary given that the 0.05 significance level is also employed. Is there a good rationale for the inclusion of the 0.1 significance level?

Response: Thank you for your comment. In this study, the Mann-Kendall test was adopted to assess the significance of climate change. However, limited by the sample length and large sample variance, the trend of climate change varies on the Tibetan Plateau. To improve the power of the Mann-Kendall test, the significance level was increased to 0.1 (Wang et al., 2020). In addition, the 0.1 significance level has been confirmed effective in the hydrological and climate trend analysis (Hamed, 2008; Hu et al., 2020; Gadedjisso-Tossou et al., 2021).

Gadedjisso-Tossou, A., Adjegan, K. I., and Kablan, A. K. M.: Rainfall and temperature trend analysis by Mann-Kendall test and significance for Rainfed Cereal Yields in Northern Togo. Sci, 3(1), 17, https://doi.org/10.3390/sci3010017, 2021.
Hamed, K. H.: Trend detection in hydrologic data: The Mann-Kendall trend test under the scaling hypothesis. J. Hydrol., 349(3-4), 350-363, https://doi.org/10.1016/j.jhydrol.2007.11.009, 2008.
Hu, Z., Liu, S., Zhong, G., Lin, H., and Zhou, Z.: Modified Mann-Kendall trend test for hydrological time series under the scaling hypothesis and its application. Hydrological Sciences Journal, 65(14), 2419-2438, https://doi.org/10.1080/02626667.2020.1810253, 2020.
Wang, F., Shao, W., Yu, H., Kan, G., He, X., Zhang, D., and Wang, G.: Re-evaluation of the power of the Mann-Kendall test for detecting monotonic trends in hydrometeorological time series. Front. Earth Sci., 8, 14, https://doi.org/10.3389/feart.2020.00014, 2020.

13. Lines 262-263: This description ('most pronounced retreat') does not appear to be entirely consistent with the panels in Fig. 4 which show rates of area reduction being greatest in zones I, III and IV, with more modest losses in zones II and VIII.

Response: Thank you for your comment. In the potential revision, this sentence will be revised as follows: Notably, the most pronounced retreat occurs at the edges, especially in Zones I and III, situated in the eastern and southeastern regions of the plateau.

14. Lines 263-264: The term 'glacier advance' is a little misleading here, given that regions VI and VIII (encompassing the Karakoram) show overall trends of area loss in Fig. 4. Presumably, 'advance' here refers to some individual glaciers that are acting counter to the overall trend in these subzones. If so, they are not easily visible in the main map in Fig. 4. Consequently, an enlarged inset of these subzones (or a separate figure) may be required to better illustrate their presence.

Response: Thank you for your kind suggestion and comment. The term of "glacier advance" refers to the area increase of some individual glaciers in these sub-zones. For illustration purposes, an enlarged inset of the glacier advance area will be displayed in the potential revision.

15. Lines 268-269: States that glacier advances are seen from 2000-2005 'in all zones except for VI and VIII'. However, zones III and IV also do not show evidence of glacier advance over this period (as illustrated in both Figs. 4 & 5).

Response: Thank you for your comment. Glacier area change in Zones I, III, and IV is minor from 2000 to 2005. To avoid this confusion, this sentence will be revised as follows: From 2000, there was a slight increase in glacier areas in Zones II, V, and VII.

16. Figure 4: To aid interpretation of the figure it would be worth noting in the figure caption that the values presented below each zone number represent the annual change in glacier area (and not the total change over the duration of the study).

Response: Thank you for your kind suggestion and comment. In the potential revision, the annual change in the glacier area will be displayed in the figure caption.

17. Lines 284-286: What might explain the much higher percentage differences between the AGEI and RGI outlines in the other zones? It is not good practice to only report the lowest differences and use them as validation of the technique.

Response: Thank you for your comment. The target year of RGI outlines is 2000, and only 65% of RGI 6.0 outlines were dated less than five years away from the target year 2000. In this study, the average glacier area from the three periods (2000, 2005, and 2010) and the RGI 6.0 outline were utilized for comparison, which will cause the percentage difference between the glacier mapping using the AGEI method and the RGI 6.0 outlines.

The quality of RGI 7.0 data is substantially improved in many regions due to the inclusion of newly updated inventory glacier data, and only 23% of all RGI 7.0 outlines were dated to five or more years away from the target year 2000 (RGI 7.0 Consortium, 2023). Thus, in the potential revision, the latest RGI 7.0 outlines will be adopted to verify the accuracy of the mapped 2000 glaciers using the AGEI method.

RGI 7.0 Consortium: Randolph Glacier Inventory - A Dataset of Global Glacier Outlines, Version 7.0 [DataSet]. Boulder, Colorado USA. NSIDC: National Snow and Ice Data Center. https://doi.org/10.5067/f6jmovy5navz, 2023.

18. Lines 290-291 & Table 2: More detail regarding these indices would be useful. My (possibly incorrect) interpretation of this table is that only around 60-70% of the RGI 6.0 glacier areas have been mapped by the AGEI technique, which seems rather low (also see Specific Comment 1 above).

Response: Thank you for your kind suggestion and comment. In this study, the reference glaciers are the RGI 6.0 outlines, and 35% of all RGI 6.0 outlines were dated to five or more years away from the target year 2000. Thus, these indices may not be higher, especially for years 2005 and 2010.

In the potential revision, the latest RGI 7.0 glacier will be adopted to validate our glacier mapping results. The validation results indicate that the accuracy of our mapped glaciers in 2000 using the latest RGI 7.0 glacier data is higher than that using the RGI 6.0 glacier data, which further confirms the accuracy of our mapped glaciers. This validation will be added in the potential revision.

19. Lines 305-306 & Figure 6 caption: more information regarding the training and validation samples would be useful. These do not appear to be mentioned in the methodology.

Response: Thank you for your kind suggestion and comment. In the potential revision, the locations of the adopted training and validation samples will be displayed in the Supplement information. In addition, the number and distribution information will be also added in the Supplement information.

20. Lines 327-329 & 337-338: It is unclear what 'glacier advance' is referring to specifically in these instances (e.g. individual glaciers in the subzones, or overall advances in some zones in particular time periods such as 2005-2010?). Also see comment regarding Lines 263-264.

Response: Thank you for your comment. The glacier advance in these two places refers to the area increase of some individual glaciers under the impacts of climate change during the observation period (i.e., from 1988 to 2022). For example, there is a notable decreasing trend in air temperature during the winter season in the transition area between Zones II and VIII, which aligns with the area increase in some individual glaciers of this region. In addition, the area of some individual glaciers increased in the southwest Himalayas and the Karakoram, which may be caused by a marginal increase in precipitation during the observation period. To avoid this confusion, the explanation of glacier advance will be added in the potential revision.

21. Lines 334-335: Avoid the use of 'respectively' in this instance (the list is too long),

and just include the name of each period (annual, fall etc.) next to each numeric value.

Response: Thank you for your kind suggestion and comment. In the potential revision, this sentence will be revised as follows: The average rates of precipitation change vary by season. They were 3.5 mm/decade annually, 7.0 mm/decade in fall, -5.0 mm/decade in winter, -1.5 mm/decade in spring, and 1.2 mm/decade in summer.

22. Figure 8: It was not immediately apparent to me that the colour key underneath the figure was universal to both sides of the figure (a & b). A note in the figure caption could help to clarify this.

Response: Thank you for your kind suggestion and comment. To avoid this confusion, the note about the colour key will be clarified, and the legend will be moved to the right place of the figure in the potential revision.

23. Lines 354-355: Presumably 'variations' here should be 'increases' to specify the direction of the relationship between glacier area and temperature? In addition, it states that 'all the subzones' have a negative correlation between glacier area and temperature change, but Figure 9a (annual) only shows zones II, III, VI and VIII to be clearly below 0 on the y-axis (thus presumably indicating glacier area loss). The coefficients and confidence intervals of the remining zones is difficult to discern. A table containing these values could be useful here (see Specific Comment 2).

Response: Thank you for your kind suggestion and comment. There is a negative correlation between glacier area changes and annual temperature increases. In the potential revision, this sentence will be revised as follows: The glacier area may decrease with the annual temperature increases.

The coefficients of the linear regression between the annual temperature and the glacier area are -42.47 $km^2/°C$ (Zone I), -1669.57 $km^2/°C$ (Zone II), -1531.32 $km^2/°C$ (Zone III), -196.12 $km^2/°C$ (Zone VI), -34.65 $km^2/°C$ (Zone V), -5762.01 $km^2/°C$ (Zone VI), -277.9 $km^2/°C$ (Zone VII), and -9341.39 $km^2/°C$ (Zone VIII). Thus, across all the sub-zones, there is a negative correlation between glacier area changes and annual temperature increases. To avoid this confusion, a detailed table containing these values will be added in the Supplement information.

24. Lines 361-362: See previous comment. A visual interpretation of Figure 9b (annual) suggests that only zones III, V, VI and VIII have confidence intervals in excess of 0 on the y-axis, and that therefore only these 4 zones can be confidently interpreted as exhibiting a positive relationship between glacier area and total precipitation at an annual scale. Equally, if the confidence intervals for zone II also cross the y-axis and it

cannot be said with confidence that the relationship is negative. I think it would be beneficial for the authors to re-write this entire paragraph with greater consideration given to their confidence intervals, particularly where they cross 0 on the y-axis.

Response: Thank you for your kind suggestion and comment. In the potential revision, the detailed table presenting the regression results will be added in the Supplement information. In the supplementary table, the regression coefficients, the 95% confidence intervals, standard error, t-values, p-values, and a summary of model diagnostics ($R^2$, Adjusted $R^2$, and RSE) will be displayed. As such, the impacts of precipitation on the glacier area change in each sub-zone will be reanalyzed.

25. Line 363: Should 'zones III, V, and VIII' read 'zones III, VI and VIII'? It also looks like zone VI could be added to this list.

Response: Thank you for your comment. The correlation coefficient of the annual precipitation and the glacier area in Zone VI is 121 $km^2$/mm. However, this coefficient of determination ($R^2$) is only 0.18. Thus, this restraining impact of annual precipitation on the glacier area is not evident in Zone VI. To avoid this confusion, this explanation will be added in the potential revision.

26. Lines 363-365 and Figure 9b: Units are inconsistent here, the text states km2/mm, but the y-axis on Figure 9b states km2/m. The abstract also states km2/mm.

Response: Thank you for your kind suggestion. The unit in Figure 9(b) is $10^3$ $km^2$/m, which is also $km^2$/mm. To avoid this confusion, the unit of Figure 9(b) will be revised as $km^2$/mm.

27. Line 383: Which earlier study; this study or Huang et al. 2021?

Response: Thank you for your comment. The study of Huang et al. (2021) is earlier. To avoid this confusion, this sentence will be revised as follows: Huang et al. (2021) utilized the minimum NDSI value at each pixel for glacier mapping, which proved to be particularly effective in minimizing the influence of seasonal snow cover.

28. Lines 405-406: For consistency, it would be useful to also include zone numbers when referring to locations in this sentence.

Response: Thank you for your kind suggestion and comment. In the potential revision, this sentence will be revised as follows: Fig. 4 highlights the distinct regional variations in glacier area changes, with the most pronounced retreat observed in the Himalayas

and the southeastern Tibetan Plateau (i.e., Zones II and III).

29. Lines 409-411: See previous comment.

Response: Thank you for your kind suggestion and comment. In the potential revision, this sentence will be revised as follows: The rising temperatures and declining precipitation in the southeastern Tibetan Plateau (i.e., Zone III), as depicted in Fig. 8, are likely contributing factors to the glacier retreat observed in this region. Moreover, a gradient of glacier shrinkage is observed, diminishing progressively from the Himalayas towards the continental interior (i.e., From Zone II to Zone V ), indicating a less pronounced retreat in these regions.

30. Lines 432-436: Given the scarcity of data points I think the authors should be cautious of giving too much weight to the data presented in Figure 11. For example, although zones I and III have lower median debris thickness and relatively high rates of glacier area loss, zone 8 has a similar level of debris thickness, but much lower rates of glacier area loss, and zone 4 has the highest median debris thickness, but still has the 3rd highest rate of glacier area loss. This paragraph could benefit from a discussion of the differences between the regional glacier area losses presented here (for debris-free ice) and any studies of regional glacier area losses for debris-covered glaciers (should the data be available).

Response: Thank you for your kind suggestion and comment. In the potential revision, the debris-covered glacier area change will be collected on the Tibetan Plateau; as such, the difference between the debris-free glacier area change and the debris-covered glacier area change can be discussed. Limited by the debris thickness data, the discussion on the influence of debris thickness on the glacier retreat will be weakened.

31. Line 451: Clarify that the 'total glacier area' is the debris-free total glacier area.

Response: Thank you for your comment. In the potential revision, this sentence will be revised as follows: The total debris-free glacier area decreased from $94.59 \times 10^3$ km$^2$ to $61.16 \times 10^3$ km$^2$, with an average annual retreat rate of $1.08 \pm 0.28 \times 10^3$ km$^2$.

32. Line 457-458: I think this sentence should be rephrased because the total glacier area in these regions has typically reduced over the duration of the study (see trends on Figure 4), rather than advanced. Given the overall reduction in glacier area and the correlations presented in Fig. 9b it may be more appropriate to state that annual and summer precipitation may have helped to mitigate reductions in glacier area in the Karakorum.

Response: Thank you for your kind suggestion and comment. In the potential revision, this sentence will be revised as follows: The annual and summer precipitation may have helped to mitigate reductions in the glacier area in the Karakorum.

33. Lines 461-464: I think this conclusion could also be revised with regard to the earlier comment about confidence intervals (see comment for Lines 361-362).

Response: Thank you for your kind suggestion and comment. In the potential revision, the impacts of annual and seasonal precipitation on the glacier area change will be reanalyzed, considering the confidence interval of the regression coefficient.

---

## Author Response (AR2)

**Response to Reviewer 1:**

The manuscript presented by Zhao and co-authors put forward a glacier mapping tool using Landsat collections and reanalysis data over the Tibetan Plateau (TP).

The manuscript is generally well written and mostly well structured, however I have made some suggestions to try and improve the flow of the manuscript.

Response: Thank you for your careful review of our manuscript and for your many high-quality comments and suggestions. We sincerely appreciate the comments that have helped sharpen this paper. Specific responses to the review comments are presented immediately after the respective review comments.

My main and minor comments can be found below:

Main:

1. Why is RGI 6.0 being used instead of RGI 7.0 which was published in September 2023? I appreciate if analysis had been carried out before the most recent release, however I think it is important the most up to date data products are used in current day studies as I know there can be large differences in glacier extents between version 6.0 and 7.0 and therefore may impact your results significantly. I will leave this to the decision of the handling editor, but I would suggest using RGI 7.0 to ensure at the time of publication, the results reflect the most current version.

Response: Thank you for your kind comment and suggestion. The RGI 6.0 was the latest glacier data when conducting this work. Thus, the validation of our glacier mapping results is mainly based on the RGI 6.0 glacier data.

The quality of RGI 7.0 data is substantially improved in many regions due to the inclusion of newly updated inventory glacier data (RGI 7.0 Consortium, 2023). Thus, in this revision, the RGI 7.0 data is adopted to validate our glacier mapping results. The validation results indicate that the accuracy of our mapped glaciers in 2000 using the latest RGI 7.0 glacier data is higher than that using the RGI 6.0 glacier data, which further confirms the accuracy of our mapped glaciers. (Please see Line 105-106, 179-181, 291-300, 305-308, 310-312, and 314-315)

2. There is no consideration of the poor 'completeness' value in Table 2, particularly for 2010. The F-1 scores for glacier mapping are moderate, I would not consider them strong metrics in support of the method. It is of course fine to have these scores, however, there needs to be consideration of why the scores are this low and how this may impact the results in the discussion.

Response: Thank you for your comment. The target year of RGI glacier data is 2000; thus, the completeness of the mapped glaciers in our study was particularly poor in 2010. This consideration was added in this revision. (Please see Line 305-308)

Further, 35% of all RGI 6.0 outlines were dated to five or more years away from the target year 2000, while this number is down to 23% in RGI 7.0 (RGI 7.0 Consortium, 2023). In addition, the

accuracy of our mapped glaciers in 2000 using the latest RGI 7.0 glacier data is higher than that using the RGI 6.0 glacier data, which further confirms the accuracy of our mapped glaciers. In this revision, the comparison results between the mapped glaciers in this study and the RGI 6.0 and RGI 7.0 glacier data were also added. (Please see Line 297-300 and 305-308)

3. Each section of the paper (i.e. results, discussion) does not need to have an introductory section of how it is structured and I would suggest deleting them and going straight into the text of the section.

Response: Thank you for your comment and suggestion. These introduction sections were deleted in this revision. (Please see Line 116, 267, and 420)

Minor:
4. L17: Would just note what time period this 'slight increase' was

Response: Thank you for your comment. Most glacier areas experienced a decrease from 1990 to 2000, followed by a slight increase from 2000 to 2010. The 'slight increase' was clarified in this revision. (Please see Line 18)

5. L19: Would consider stating geographically where these 'zones' are as out of context as they don't mean very much as no lead in from the abstract (i.e. NE of TP)

Response: Thank you for your kind suggestion. The southern Himalayas replaced Zone VIII, and the Karakoram replaced Zone VI in this revision. (Please see Line 20 and 21)

6. L35: I get what you mean, but I would rephrase the opening on this paragraph saying 'Glacier change can be measured using variations in...' to be clearer

Response: Thank you for your kind suggestion. This sentence was revised as follows: Glacier change can be measured using variations in area, thickness, volume, surface mass balance, and equilibrium line altitude. (Please see Line 36-37)

7. L42: Would put the reference for each method after it was referred to i.e., spectral analysis (ref), object-based (ref), etc.

Response: Thank you for your suggestion and comment. This sentence was revised as follows: Glaciers are commonly mapped using a variety of techniques, including spectral analysis of optical satellite images (Bolch et al., 2010), object segmentation-based methods (Robson et al., 2015), and supervised machine-learning algorithms (Khan et al., 2020). (Please see Line 42-44)

8. L46: needs a reference for debris-free glaciers before full stop

Response: Thank you for your kind comment. The reference for debris-free glaciers was added as follows: Robust and efficient pixel-based multispectral analysis has been particularly effective in accurately delineating debris-free glaciers (Huang et al., 2021). (Please see Line 47)

9. L51: are 'quality' and 'resolution' not the same thing? Also, what type of resolution, spatial or temporal? If both I would suggest stating the limitation of spatio-temporal resolution over the TP.

Response: Thank you for your comment and suggestion. There are some differences between quality and resolution; specifically, the resolution is one type of quality. In this study, they both indicate the spatial and temporal resolutions of the glacier mapping on the Tibetan Plateau. However, glacier mapping on the Tibetan Plateau with high spatial-temporal resolutions is limited due to the large amounts of satellite images and the massive computing. To avoid this confusion, this sentence was revised as follows: Furthermore, limited by the large amounts of satellite images and massive computing, the comprehensive depiction of glacier retreat across the entire Tibetan Plateau, especially at finer temporal and spatial resolutions, remains inadequately characterized. (Please see Line 50-52)

10. L58: Climate change seems a bit broad here, is it the increasing air temperature? Would clarify

Response: Thank you for your comment and suggestion. The dominant driver of the glacier retreat is the increasing temperature. To avoid this confusion, this sentence was revised as follows: Climate change, especially the increasing temperature, is recognized to be the dominant driver of the glacier mass balance and the associated area and volume changes. (Please see Line 58)

11. L77: If you state 'numerous studies' I would suggest citing a handful of them as examples

Response: Thank you for your kind suggestion. More references were added in this revision, and the revised sentence was as follows: Despite numerous studies examining glacier variations on the Tibetan Plateau in recent decades (Yao et al., 2012; Neckel et al., 2014; Ye et al., 2017; Bibi et al., 2018; Sun et al., 2018; Latif et al., 2019; Zhang et al., 2021; Xiao et al., 2023), the specific impacts of climate change on glacier retreat have not been thoroughly investigated at a finer resolution. (Please see Line77-78)

12. L82-85: not convinced this is required for such a specific structure, think you're fine with just the aims of the paper being highlighted

Response: Thank you for your kind comment and suggestion. These sentences are intended to understand the structure of this article. To avoid this confusion, these sentences were deleted in this

revision. (Please see Line 83)

13. L86: Just call it Study Area

Response: Thank you for your comment. This subheading was revised as 'Study Area'. (Please see Line 84)

14. L117: Would call it data and methods instead of materials

Response: Thank you for your comment. This subheading was revised as 'Data and Methods'. (Please see Line 115)

15. L118: This text is not needed - would go straight to 3.1

Response: Thank you for your comment. In this revision, this text was deleted. (Please see Line 116)

16. L124: I would just be careful saying images via GEE catalog are 'open-access' - while they are for individuals, there is commercial cost to access the platform

Response: Thank you for your comment and suggestion. In this revision, the 'open-access' was deleted to avoid this confusion, and the revised sentence was as follows: The Landsat data is used via the Google Earth Engine platform, which is attributed to their prolonged data availability period and comparatively high spatial resolution. (Please see Line 119-120)

17. L124: Would merge these sentences, suggest at start of sentence saying Landsat data is used via GEE

Response: Thank you for your comment and suggestion. In this revision, this sentence was revised as follows: The Landsat data is used via the Google Earth Engine platform, which is attributed to their prolonged data availability period and comparatively high spatial resolution. (Please see Line 119-120)

18. L158: Figure 2 caption - Is this the total number of Landsat images or the total number used in the study with less than 60% cloud as defined by your study? Would clarify

Response: Thank you for your comment. The number of Landsat images is the total number used in the study with less than 60% cloud. In this revision, this sentence was revised as follows: The number of Landsat images with less than 60% cloud available for each year corresponding to each

period of glacier mapping is depicted in Fig. 2. In addition, the caption of Fig. 2 was revised as follows: Number of Landsat images with less than 60% cloud for each year corresponding to each period of glacier mapping. (Please see Line 142-143 and 151)

19. L162-164: basically, the same as previous, would suggest merging and stating previous studies chose 0.4 as a threshold and therefore it was chosen here

Response: Thank you for your kind comment and suggestion. In this revision, this sentence was revised as follows: Previous studies chose 0.4 as a threshold to extract snow and ice (Scherler et al., 2018; Huang et al., 2021); thus, in this study, a threshold value of 0.4 is set to facilitate the extraction of debris-free glaciers from the Landsat images. (Please see Line 156-158)

20. L167-169: same applies about thresholding values NDWI, just merge them and say you chose 0.4

Response: Thank you for your kind comment and suggestion. In this revision, this sentence was revised as follows: Many studies have depicted that the NDWI values of the water pixels ranged from 0.4 to 1 (Du et al., 2016; Zhao et al., 2018; Bevington and Menounos, 2022); thus, in this study, an NDWI threshold of 0.4 is adopted to minimize errors associated with the presence of open water in the glacier mapping. (Please see Line 162-164)

21. L171: "Therefore" does not fit here as does not follow the previous sentence

Response: Thank you for your comment. In this revision, this sentence was revised as follows: Based on prior research and preliminary analysis of the surface temperature of the reference RGI 7.0 glaciers, a threshold for surface temperature (derived from the thermal band) is set at -1 °C (Shugar et al., 2020). (Please see Line 165-167)

22. L173-174: the final paragraph is just repetition of values you've defined - suggest deleting

Response: Thank you for your kind suggestion. To avoid this repetition, this sentence was deleted in this revision. (Please see Line 167)

23. L183: What holes? What do you mean "filled"? Interpolated? Would clarify

Response: Thank you for your comment. These holes may be a part of the mapped individual glaciers. Limited by the spatial resolution of the used Landsat images and the processing errors, there may be some holes in the mapped individual glacier. In this study, these holes smaller than 0.01 km² are filled according to previous studies (Bevington and Menounos, 2022); as such, a more

complete glacier mapping can be obtained. To avoid this confusion, this sentence was revised as follows: Limited by the spatial resolution of the used Landsat images, there may be some errors and holes in the mapped individual glacier. In this study, polygons with an area less than 0.05 km² are excluded, and holes smaller than 0.01 km² are filled according to previous studies (Bevington and Menounos, 2022). (Please see Line 174-177)

24. L185: sentence does not make sense - what do you mean 'validated using reference debris-free glaciers'?

Response: Thank you for your comment. The mapped glaciers in this study are debris-free glaciers; thus, the accuracy of the mapped debris-free glaciers is validated using reference RGI 6.0 and RGI 7.0 debris-free glaciers. These reference glaciers are derived by removing the debris-covered portions from the RGI 6.0 and 7.0 datasets, with the debris regions sourced from Scherler et al. (2018). In this revision, this sentence was revised as follows: The mapped glaciers in this study are debris-free glaciers; thus, the accuracy of the mapped debris-free glaciers is validated using reference RGI 6.0 and RGI 7.0 debris-free glaciers. These reference glaciers are derived by removing the debris-covered portions from the RGI 6.0 and RGI 7.0 datasets, with the debris regions sourced from Scherler et al. (2018). (Please see Line 179-181)

25. L253: Can Fig 3 be made bigger?

Response: Thank you for your comment and suggestion. In this revision, Fig. 3 was made bigger. (Please see Line 264)

26. L255-258: Delete introductory

Response: Thank you for your comment. In this revision, this paragraph was deleted. (Please see Line 267)

27. L275: Figure 4. Would suggest having the map as a) at the top of the figure then having panels b-i stacked. The lettering order seems a little confusing

Response: Thank you for your comment and suggestion. In this revision, the order was re-sorted alphabetically. (Please see Line 285-289)

28. L287: What do you mean the glaciers mapped between 2000 and 2005 "exhibit greater consistency"?

Response: Thank you for your comment. Fig. 5 depicts the comparison results of glacier mapping

in 2000, 2005, and 2010 and the reference RGI 6.0 and RGI 7.0 debris-free glaciers. It describes that the glaciers mapped in 2000 and 2005 exhibit greater consistency compared to those in 2010, with this trend being particularly pronounced in Zone III. In this revision, the latest RGI 7.0 was used as the reference glaciers, and the comparisons between the mapped glaciers in this study and the RGI 7.0 glaciers were depicted. (Please see Line 297-300 and 305-308)

29. L300: Table 2 metric scores - see main comment 2

Response: Thank you for your comment. The target year of RGI glacier data is 2000; thus, the completeness of the mapped glaciers in our study was particularly poor in 2010. This consideration was added in this revision. Further, 35% of all RGI 6.0 outlines were dated to five or more years away from the target year 2000, while this number is down to 23% in RGI 7.0 (RGI 7.0 Consortium, 2023). Thus, the accuracy of our mapped glaciers in 2000 using the latest RGI 7.0 glacier data is higher than that using the RGI 6.0 glacier data, which further confirms the accuracy of our mapped glaciers. In this revision, the comparison results between the mapped glaciers in this study and the RGI 6.0 and RGI 7.0 glacier data were also added. (Please see Line 297-300 and 305-308)

30. L400: I think Fig.10 and above text is more results than discussion. Would suggest moving to results. Do you have any numbers for the total area difference between the two methods? Would add to the argument of the AGEI method

Response: Thank you for your comment and suggestion. In this revision, the comparison results of the mapped glacier using the AGEI method and the minimum NDSI method were moved to the results section, and the quantitative comparison between these two methods was added in the results section. (Please see Line 317-334)

31. L405: Would refrain from starting discussion sentences 'Fig X', if wanting to directly refer to the figure, place in brackets at the end of the sentence

Response: Thank you for your kind suggestion. In this revision, this sentence was revised as follows: The distinct regional variations in changes of glacier extent, with the most pronounced retreat observed in the Himalayas and the southeastern Tibetan Plateau (i.e., Zones II and III), as shown in Fig. 4. (Please see Line 421-422)

**Response to Reviewer 2:**

GENERAL COMMENTS

This manuscript investigates how changes in debris-free glacier extent in the Tibetan Plateau between ~1990-2020 are related to changes in climate (temperature, precipitation) over the same period. The authors: i) outline a novel technique (the AGEI) for mapping glacier extent, and use it to quantify changes in glacier area at 5-year intervals for 8 subzones of the Tibetan Plateau; ii) downscale ERA5-Land reanalysis data (specifically 2 m temperature and total precipitation) from 9 $km^2$ to 1 $km^2$ using ancillary variables from MODIS and the SRTM DEM) in a random forest regression, and; iii) use linear regressions to investigate the relationships between changes in glacier extent, temperature and precipitation at annual and seasonal scales. The authors conclude that total (debris-free) glacier area in the region has reduced over the survey duration, that temperatures have typically risen, that precipitation has increased in some subzones and reduced in others, and that in some subzones (particularly those in the Karakorum) reductions in glacier area correlated strongly with increases in temperature, but that the severity of glacier area loss may have been mitigated by corresponding increases in precipitation.

The manuscript presents a significant body of work, including novel mapping methodologies, extensive glacier and climate datasets, and insights into regional glacier and climate change in the Tibetan Plateau, and should therefore be of interest to others mapping glacier change generally, and particularly researchers investigating glacier change in central Asia. The addition of a paragraph discussing the implications of the glacier - climate relationships identified in this work with regard to existing climate projections for the region could help add impact. The manuscript is largely well-written, structured and presented, and could be further improved with some edits to make it more succinct (see comments for specific guidance).

My main comments relate to the methods adopted, several of which require additional clarification and/or detail to by fully comprehensible and transparent. I think this is of particular importance given the novelty of the mapping method used and the borderline nature of some of the glacier - climate relationships presented (e.g. see confidence intervals in Fig. 9). A caveat to my comments is that I am not an expert in Himalayan/Tibetan glaciation, or well-versed in random forest or time-series analyses, and have thus taken the methods adopted in section 3.2 (specifically lines 211-250) in good faith.

Response: Thank you for your careful review of our manuscript and for your many high-quality comments and suggestions. We sincerely appreciate the comments and suggestions that will help sharpen this paper. In this revision, the sixth phase of the coupled model intercomparison project (CMIP6) was adopted to assess future climate change, and three scenarios (SSP1-2.6, SSP2-4.5, and SSP5-8.5) were used for future climate projections, as such, the future glacier area change was discussed under different climate change scenarios. (Please see Line 465-485)

Further, the linear regression method was also clarified in this revision, and the detailed results of the regression (i.e., Fig. 9) were presented in the supplement. Then, the impact of the annual and

seasonal temperature and precipitation change on the glacier area change in different sub-zones of the Tibetan Plateau was re-discussed. Specific responses to the review comments are given immediately after the respective review comments. (Please see Line 252-262 and Tables S1 and 2)

SPECIFIC COMMENTS

1. The accuracy and consistency of the novel AGEI mapping technique is integral to the results of this analysis. Consequently, a more detailed evaluation of the technique would be welcome, both to aid replication and assess its effectiveness. In particular, Fig. 10 does not have sufficient resolution to assess the accuracy of the AGEI mapping. Instead, a figure with multiple high-resolution panels (including some features that present challenges to accurate glacier mapping, such as lakes, shadow, cloud etc.), would be useful to illustrate the accuracy of the AGEI glacier outlines, particularly if the reference RGI 6.0 glacier outlines can be overlaid on them. Given its importance to the results, I think such a figure would be better situated and discussed in Section 3.1 or Supplementary Information, rather than waiting until the discussion.

In addition, I struggled to fully understand the mapping evaluation indices presented on lines 189-198, particularly the Correctness (differentiating between Acg and Atg) and F1 scores (e.g. what does a high F1 score indicate?). If possible, illustrations of high and low scoring glaciers/regions/test sites (e.g. in Supplementary Information) could aid interpretation of the indices and the demonstrate the accuracy of the AGEI technique.

Response: Thank you for your kind comment and suggestion. In this revision, a detailed evaluation of the AGEI method was added in the results section. Particularly, the section of discussion on the proposed AGEI method was moved to the results part, and a figure with multiple high-resolution panels (including some features that present challenges to accurate glacier mapping, such as lakes, shadows, clouds, etc.) was added to illustrate the accuracy of the AGEI method. (Please see Line 317-340)

Further, the three mapping indices (Correctness, Completeness, and F1-score) were described in Section 3.1. For example, Correctness is computed as the ratio of the correctly mapped glacier area to the total mapped glacier area in this study, and the F1 score provides a balance between Completeness and Correctness, and the high F1 score indicates the high accuracy of our mapped glaciers when using the RGI glacier as the reference glacier. For illustration purposes, a glacier test site was also added in the supplement. (Please see Line 183-195 and Figure S1)

2. An additional section should be added to the end of the methodology to describe the linear regressions used to investigate the relationships between temperature/precipitation and glacier change (the linear model is mentioned at the beginning of Section 4.3, but not described fully).

For transparency it would be useful to write out the model equation(s) used in the methods. It is unclear whether numerous independent models were generated (e.g. one for each for each climate variable, zone and season, totalling ~80 independent regressions), or if these were all included in

one multiple regression model (and, if so, how multicollinearity between the seasonal climate was addressed).

The results of the regressions should also be presented in a table that includes the coefficients, confidence intervals, standard error, t-values and p-values, and a summary of model diagnostics (R2/Adjusted R2, RSE etc.). Although the coefficients and confidence intervals are presented in Fig. 9 their numeric values are difficult to determine, hence a table of the regression outputs would be useful to support this figure, aid interpretation of the data, and provide more robust conclusions.

Response: Thank you for your kind comment and suggestion. In this revision, the linear regression method adopted was detailed at the end of Section 3. (Please see Line 252-262)

In this study, the impacts of annual and seasonal temperature and precipitation on the glacier area change are included in one regression model for simplified display. Though studies showed that there is a positive scaling between extreme precipitation and surface air temperature (Yong et al., 2021), the relationship between precipitation and temperature is complex due to the unique geographical and climatic features, and the correlation between the temperature and precipitation is not significant, particularly on an annual or seasonal basis (Duan and Xiao, 2015; Wu et al., 2015). In addition, several studies have proved the effectiveness of the separate impacts of temperature and precipitation on the glacier retreat (Li et al., 2019; Bevington and Menounos, 2022). Thus, the interactions between air temperature and precipitation were not accounted for in this analysis. In the future study, the interaction between air temperature and precipitation will be considered in analyzing the impacts of climate change on the glacier retreat. (Please see Line 260-262)

In this revision, the table presenting the regression results was added in the supplement. In Tables S1 and S2, the coefficients, confidence intervals, standard error, t-values, p-values, and a summary of model diagnostics ($R^2$, Adjusted $R^2$, and RSE) were displayed. (Please see Tables S1 and S2)

Bevington, A. R., and Menounos, B.: Accelerated change in the glaciated environments of western Canada revealed through trend analysis of optical satellite imagery. Remote Sens Environ., 270, 112862, https://doi.org/10.1016/j.rse.2021.112862, 2022.

Duan, A., and Xiao, Z.: Does the climate warming hiatus exist over the Tibetan Plateau? Sci Rep., 5(1), 13711, https://doi.org/10.1038/srep13711, 2015.

Li, Y. J., Ding, Y. J., Shangguan, D. H., and Wang, R. J.: Regional differences in global glacier retreat from 1980 to 2015. Adv. Clim. Change Res., 10(4), 203-213, https://doi.org/10.1016/j.accre.2020.03.003, 2019.

Wu, G., Duan, A., Liu, Y., Mao, J., Ren, R., Bao, Q., and Hu, W.: Tibetan Plateau climate dynamics: recent research progress and outlook. Natl Sci Rev., 2(1), 100-116, https://doi.org/10.1093/nsr/nwu045, 2015.

Yong, Z., Xiong, J., Wang, Z., Cheng, W., Yang, J., and Pang, Q.: Relationship of extreme precipitation, surface air temperature, and dew point temperature across the Tibetan Plateau. Clim Change., 165, 41, https://doi.org/10.1007/s10584-021-03076-2, 2021.

3. A major caveat of this work is that the analysis presented only considers debris-free glaciers. Greater context regarding the prevalence and distribution of debris-covered glaciers in the region would therefore be useful (in the introduction, or sections 3.1 or 5.3) to understand the potential implications of this methodological choice on the results and their interpretation (e.g. are debris-free glaciers typically representative of glaciers across the region; could there be some sub-zones where the exclusion of debris-covered glaciers from the analysis significantly reduces the sample size and therefore the representativeness of the results?).

Response: Thank you for your kind comment. Using the glacier boundaries of RGI, it was estimated that 10% of glaciers are covered by debris on the Tibetan Plateau and its surrounding areas (Scherler et al. 2018), and the debris-covered glaciers on the Tibetan Plateau are mainly distributed in the Himalayas (Ojha et al., 2017; Ji et al., 2022). The debris layer makes the melting process of glaciers complicated, and the debris-covered glaciers have contrasting melting mechanisms and climate response patterns if compared with debris-free glaciers (Chen et al., 2023; He et al., 2023). Thus, the impacts of climate change on the debris-free and debris-covered glaciers retreats are usually analyzed separately (Hu et al., 2022).

However, it is difficult to recognize debris automatically from satellite images. Only when glacier boundaries are given can the discrimination of debris from snow and ice become operational (Huang et al., 2022). Considering that there is no accurate method to extract debris change, thus, in this study, glacier area change only refers to changes in debris-free glaciers, and the exclusion of debris-covered glaciers from the analysis will not reduce the sample size. To avoid this confusion, the distribution and discussion of debris-covered glaciers were added in this revision. (Please see Line 120-122 and 440-444)

Chen, F., Wang, J., Li, B., Yang, A., and Zhang, M.: Spatial variability in melting on Himalayan debris-covered glaciers from 2000 to 2013. Remote Sens Environ., 291, 113560, https://doi.org/10.1016/j.rse.2023.113560, 2023.

He, Z., Yang, W., Wang, Y., Zhao, C., Ren, S., and Li, C.: Dynamic changes of a thick debris-covered glacier in the southeastern Tibetan Plateau. Remote Sens., 15(2), 357, https://doi.org/10.3390/rs15020357, 2023.

Hu, M., Zhou, G., Lv, X., Zhou, L., Wang, X., He, X., and Tian, Z.: Warming Has Accelerated the Melting of Glaciers on the Tibetan Plateau, but the Debris-Covered Glaciers Are Rapidly Expanding. Remote Sens., 15(1), 132, https://doi.org/10.3390/rs15010132, 2022.

Huang, L., Li, Z., Zhou, J. M., and Zhang, P.: An automatic method for clean glacier and nonseasonal snow area change estimation in High Mountain Asia from 1990 to 2018. Remote Sens Environ., 258, 112376, https://doi.org/10.1016/j.rse.2021.112376, 2021.

Ji, Q., Yang, T. B., Li, M. Q., Dong, J., Qin, Y., and Liu, R.: Variations in glacier coverage in the Himalayas based on optical satellite data over the past 25 years. Catena, 214, 106240, https://doi.org/10.1016/j.catena.2022.106240, 2022.

Ojha, S., Fujita, K., Sakai, A., Nagai, H., and Lamsal, D.: Topographic controls on the debris-cover extent of glaciers in the Eastern Himalayas: Regional analysis using a novel high-resolution glacier inventory. Quat. Int., 455, 82-92, https://doi.org/10.1016/j.quaint.2017.08.007, 2017.

Scherler, D., Wulf, H., and Gorelick, N.: Global assessment of supraglacial debris-cover extents.

Geophys Res Lett., 45(21), 11-798, https://doi.org/10.1029/2018GL080158, 2018.

4. Given the relationships identified between changes in glacier area, temperature and precipitation from ~1990-2020, I think a paragraph could be added to the discussion to explore the implications of these results in respect of existing climate projections for the region. For example, how is regional climate likely to change by the end of the century and what could be the implications for the glaciers in the different sub-zones given the trends identified between 1990-2020? A short paragraph addressing this could help to add impact to the manuscript, and potentially identify future research priorities.

Response: Thank you for your kind comment and suggestion. In this revision, the sixth phase of the coupled model intercomparison project (CMIP6) was adopted to assess future climate change, and three scenarios (SSP1-2.6, SSP2-4.5, and SSP5-8.5) were used for future climate projections. As such, the average temperature and precipitation change in the eight sub-zones of the Tibetan Plateau was displayed by the end of the century, and the future glacier area change under three climate change scenarios was discussed. (Please see Line 465-485)

5. Throughout the manuscript the authors refer to 'glacier retreat' which (to me at least) implies a measure of distance retreated by a glacier terminus, particularly the terminus of a valley or outlet glacier. Given that the work presented is concerned with changes in glacier area (not just recession at termini)' glacier extent' may be a more intuitive term to use.

Response: Thank you for your kind comment and suggestion. This study mainly analyzed the glacier area change on the Tibetan Plateau, which is also the change in glacier extent. However, Glacier retreat can be measured using reductions in area, thickness, volume, surface mass balance, and increase of equilibrium line altitude (Sugiyama et al., 2013; Su et al., 2022). The retreat of glaciers located on the Tibetan Plateau was characterized by severe area shrinkage and mass loss (Li et al., 2019). In addition, many studies have depicted glacier retreat using glacier area change (Dyurgerov et al., 2009; Li et al., 2019; Sommer et al., 2020). To avoid this confusion, the glacier area change was described as the change of glacier extent, and the glacier retreat was used in some discussion parts. (Please see Line 15, 79, 139, 140, 270, 276, 382, 394, 402, 412, 418, and 499)

Dyurgerov, M., Meier, M. F., and Bahr, D. B.: A new index of glacier area change: a tool for glacier monitoring. J Glaciol., 55(192), 710-716, https://doi.org/10.3189/002214309789471030, 2009.
Li, Y. J., Ding, Y. J., Shangguan, D. H., and Wang, R. J.: Regional differences in global glacier retreat from 1980 to 2015. Adv. Clim. Change Res., 10(4), 203-213, https://doi.org/10.1016/j.accre.2020.03.003, 2019.
Sommer, C., Malz, P., Seehaus, T. C., Lippl, S., Zemp, M., and Braun, M. H.: Rapid glacier retreat and downwasting throughout the European Alps in the early 21st century. Nat Commun, 11(1), 3209, https://doi.org/10.1038/s41467-020-16818-0, 2020.
Su, B., Xiao, C., Chen, D., Huang, Y., Che, Y., and Zhao, H.: Glacier change in China over past decades: Spatiotemporal patterns and influencing factors. Earth-Sci. Rev., 226, 103926,

https://doi.org/10.1016/j.earscirev.2022.103926, 2022.

Sugiyama, S., Fukui, K., Fujita, K., Tone, K., and Yamaguchi, S.: Changes in ice thickness and flow velocity of Yala Glacier, Langtang Himal, Nepal, from 1982 to 2009. Ann. Glaciol., 54(64), 157-162, https://doi.org/10.3189/2013AoG64A111, 2013.

6. The manuscript could be made more succinct by removing several sections of text outlining the structure (which is self-evident), particularly lines 82-85, 118-119, 255-257, and 380-381.

Response: Thank you for your kind comment and suggestion. These introduction sections were deleted in this revision. (Please see Line 83, 116, 267, and 420)

7. Figures 4 & 5: Both figures show an increase in glacier area in zones II, V and VII between 2000 and 2005. This seems unusual given the overall regional trend, so I'm interested to know whether this is an artefact of the mapping methodology, or whether this brief increase in glacier extent in these regions has been documented in any other studies/literature? The same question also applies to zones VI and VIII, which appear to show notable increases in glacier area between 2005-2010.

Response: Thank you for your comment. Such glacier area increases are also documented in the references. For example, some studies show that there is a slight increase in glacier area in East Kunlun, Inner Tibet, and Central Himalayas from 2000 to 2005 (Huang et al., 2021); these areas correspond to Zones II and V. In addition, some studies show that the glacier area in the Karakoram (i.e., Zones VI and VIII) increased from 2006 to 2010 (Yao et al., 2012). To avoid this confusion, the glacier area increase in these zones was detailed in this revision. (Please see Line 279-283)

Huang, L., Li, Z., Zhou, J. M., and Zhang, P.: An automatic method for clean glacier and nonseasonal snow area change estimation in High Mountain Asia from 1990 to 2018. Remote Sens. Environ., 258, 112376, https://doi.org/10.1016/j.rse.2021.112376, 2021.

Yao, T., Thompson, L., Yang, W., Yu, W., Gao, Y., Guo, X., and Joswiak, D.: Different glacier status with atmospheric circulations in Tibetan Plateau and surroundings. Nature Clim Change, 2(9), 663-667, https://doi.org/10.1038/nclimate1580, 2012.

TECHNICAL COMMENTS

10. Abstract: Here, references to the zones used in the study would be better substituted with the names of the corresponding regions (e.g. Karakorum, southern Himalaya etc.) because the zones are only intelligible to those who have already read Section 2.

Response: Thank you for your kind comment and suggestion. The southern Himalayas replaced Zone VIII, and the Karakoram replaced Zone VI in this revision. (Please see Line 20 and 21)

11. Lines 209-210: Is there a precedent or justification for the use of the 3 ancillary MODIS and

SRTM DEM variables to downscale the ERA5-Land data?

Response: Thank you for your comment. Yes, the ERA5-Land is commonly downscaled based on ancillary factors, these ancillary factors include MODIS surface reflectances, NDVI, and DEM data (Kusch and Davy, 2022; Karaman and Akyürek, 2023; Wang et al., 2023). In this revision, more references were cited to support the downscaling analysis. (Please see Line 207-208)

Karaman, Ç. H., and Akyürek, Z.: Evaluation of near-surface air temperature reanalysis datasets and downscaling with machine learning based Random Forest method for complex terrain of Turkey. Adv. Space Res., 71(12), 5256-5281, https://doi.org/10.1016/j.asr.2023.02.006, 2023.
Kusch, E., and Davy, R.: KrigR-a tool for downloading and statistically downscaling climate reanalysis data. Environ. Res. Lett., 17(2), 024005, https://doi.org/10.1088/1748-9326/ac48b3, 2022.
Wang, N., Tian, J., Su, S., and Tian, Q.: A Downscaling Method Based on MODIS Product for Hourly ERA5 Reanalysis of Land Surface Temperature. Remote Sens., 15(18), 4441, https://doi.org/10.3390/rs15184441, 2023.

12. Line 247: A significance level of 0.1 seems unusually low, and unnecessary given that the 0.05 significance level is also employed. Is there a good rationale for the inclusion of the 0.1 significance level?

Response: Thank you for your comment. In this study, the Mann-Kendall test was adopted to assess the significance of climate change. However, limited by the sample length and large sample variance, the trend of climate change varies on the Tibetan Plateau. To improve the power of the Mann-Kendall test, the significance level was increased to 0.1 (Wang et al., 2020). In addition, the 0.1 significance level has been confirmed effective in the hydrological and climate trend analysis (Hamed, 2008; Hu et al., 2020; Gadedjisso-Tossou et al., 2021). To avoid this confusion, the explanation on the adopted significance level was added in this revision. (Please see Line 247-250)

Gadedjisso-Tossou, A., Adjegan, K. I., and Kablan, A. K. M.: Rainfall and temperature trend analysis by Mann-Kendall test and significance for Rainfed Cereal Yields in Northern Togo. Sci, 3(1), 17, https://doi.org/10.3390/sci3010017, 2021.
Hamed, K. H.: Trend detection in hydrologic data: The Mann-Kendall trend test under the scaling hypothesis. J. Hydrol., 349(3-4), 350-363, https://doi.org/10.1016/j.jhydrol.2007.11.009, 2008.
Hu, Z., Liu, S., Zhong, G., Lin, H., and Zhou, Z.: Modified Mann-Kendall trend test for hydrological time series under the scaling hypothesis and its application. Hydrological Sciences Journal, 65(14), 2419-2438, https://doi.org/10.1080/02626667.2020.1810253, 2020.
Wang, F., Shao, W., Yu, H., Kan, G., He, X., Zhang, D., and Wang, G.: Re-evaluation of the power of the Mann-Kendall test for detecting monotonic trends in hydrometeorological time series. Front. Earth Sci., 8, 14, https://doi.org/10.3389/feart.2020.00014, 2020.

13. Lines 262-263: This description ('most pronounced retreat') does not appear to be entirely consistent with the panels in Fig. 4 which show rates of area reduction being greatest in zones I, III

and IV, with more modest losses in zones II and VIII.

Response: Thank you for your comment. In this revision, this sentence was revised as follows: Notably, the most pronounced retreat occurs at the edges, especially in Zones I and III, situated in the eastern and southeastern regions of the plateau. (Please see Line 271-272)

14. Lines 263-264: The term' glacier advance' is a little misleading here, given that regions VI and VIII (encompassing the Karakoram) show overall trends of area loss in Fig. 4. Presumably, 'advance' here refers to some individual glaciers that are acting counter to the overall trend in these subzones. If so, they are not easily visible in the main map in Fig. 4. Consequently, an enlarged inset of these subzones (or a separate figure) may be required to better illustrate their presence.

Response: Thank you for your kind suggestion and comment. The term 'glacier advance' refers to the area increase of some individual glaciers in these sub-zones. For illustration purposes, an enlarged inset of the glacier advance area was displayed in this revision. (Please see Line 272-273 and Fig. 4a)

15. Lines 268-269: States that glacier advances are seen from 2000-2005 'in all zones except for VI and VIII'. However, zones III and IV also do not show evidence of glacier advance over this period (as illustrated in both Figs. 4 & 5).

Response: Thank you for your comment. Glacier area change in Zones I, III, and IV is minor from 2000 to 2005. To avoid this confusion, this sentence was revised as follows: From 2000, there was a slight increase in glacier areas in Zones II, V, and VII. (Please see Line 277-278)

16. Figure 4: To aid interpretation of the figure it would be worth noting in the figure caption that the values presented below each zone number represent the annual change in glacier area (and not the total change over the duration of the study).

Response: Thank you for your kind suggestion and comment. In this revision, the annual change in the glacier area was displayed in the figure caption. (Please see Fig. 4)

17. Lines 284-286: What might explain the much higher percentage differences between the AGEI and RGI outlines in the other zones? It is not good practice to only report the lowest differences and use them as validation of the technique.

Response: Thank you for your comment. The target year of RGI outlines is 2000, and only 65% of RGI 6.0 outlines were dated less than five years away from the target year 2000. In this study, the average glacier area from the three periods (2000, 2005, and 2010) and the RGI 6.0 outline were utilized for comparison, which might cause the percentage difference between the glacier mapping

using the AGEI method and the RGI 6.0 outlines.

The quality of RGI 7.0 data is substantially improved in many regions due to the inclusion of newly updated inventory glacier data, and only 23% of all RGI 7.0 outlines were dated to five or more years away from the target year 2000 (RGI 7.0 Consortium, 2023). Thus, in this revision, the latest RGI 7.0 outlines were adopted to verify the accuracy of the mapped 2000 glaciers using the AGEI method. (Please see Line 291-315)

RGI 7.0 Consortium: Randolph Glacier Inventory - A Dataset of Global Glacier Outlines, Version 7.0 [DataSet]. Boulder, Colorado USA. NSIDC: National Snow and Ice Data Center. https://doi.org/10.5067/f6jmovy5navz, 2023.

18. Lines 290-291 & Table 2: More detail regarding these indices would be useful. My (possibly incorrect) interpretation of this table is that only around 60-70% of the RGI 6.0 glacier areas have been mapped by the AGEI technique, which seems rather low (also see Specific Comment 1 above).

Response: Thank you for your kind suggestion and comment. In this study, the reference glaciers are the RGI 6.0 outlines, and 35% of all RGI 6.0 outlines were dated to five or more years away from the target year 2000. Thus, these indices may not be higher, especially for years 2005 and 2010.

In this revision, the latest RGI 7.0 glacier was adopted to validate our glacier mapping results. The validation results indicate that the accuracy of our mapped glaciers in 2000 using the latest RGI 7.0 glacier data is higher than that using the RGI 6.0 glacier data, which further confirms the accuracy of our mapped glaciers. This validation was added in this revision. (Please see Line 297-300 and 305-308)

19. Lines 305-306 & Figure 6 caption: more information regarding the training and validation samples would be useful. These do not appear to be mentioned in the methodology.

Response: Thank you for your kind suggestion and comment. In this revision, the distribution of the adopted training and validation samples was displayed in the supplement. (Please see Line 346-348 and Figure S2)

20. Lines 327-329 & 337-338: It is unclear what 'glacier advance' is referring to specifically in these instances (e.g. individual glaciers in the subzones, or overall advances in some zones in particular time periods such as 2005-2010?). Also see comment regarding Lines 263-264.

Response: Thank you for your comment. The glacier advance in these two places refers to the area increase of some individual glaciers under the impacts of climate change during the observation period (i.e., from 1988 to 2022). For example, there is a notable decreasing trend in air temperature during the winter season in the transition area between Zones II and VIII, which aligns with the area increase in some individual glaciers of this region. In addition, the area of some individual glaciers

increased in the southwest Himalayas and the Karakoram, which may be caused by a marginal increase in precipitation during the observation period. To avoid this confusion, the explanation of glacier advance was added in this revision. (Please see Line 370-372 and 379-380)

21. Lines 334-335: Avoid the use of 'respectively' in this instance (the list is too long), and just include the name of each period (annual, fall etc.) next to each numeric value.

Response: Thank you for your kind suggestion and comment. In this revision, this sentence was revised as follows: The average rates of precipitation change vary by season. They were 3.5 mm/decade annually, 7.0 mm/decade in fall, -5.0 mm/decade in winter, -1.5 mm/decade in spring, and 1.2 mm/decade in summer. (Please see Line 377-378)

22. Figure 8: It was not immediately apparent to me that the colour key underneath the figure was universal to both sides of the figure (a & b). A note in the figure caption could help to clarify this.

Response: Thank you for your kind suggestion and comment. To avoid this confusion, the note about the color key has been clarified, and the legend was moved to the right place of the figure in this revision. (Please see Line 389-391)

23. Lines 354-355: Presumably 'variations' here should be 'increases' to specify the direction of the relationship between glacier area and temperature? In addition, it states that 'all the subzones' have a negative correlation between glacier area and temperature change, but Figure 9a (annual) only shows zones II, III, VI and VIII to be clearly below 0 on the y-axis (thus presumably indicating glacier area loss). The coefficients and confidence intervals of the remining zones is difficult to discern. A table containing these values could be useful here (see Specific Comment 2).

Response: Thank you for your kind suggestion and comment. There is a negative correlation between glacier area changes and annual temperature increases. In this revision, this sentence was revised as follows: Across all the sub-zones, the glacier area may decrease with the annual temperature increases. (Please see Line 395-396)

The coefficients of the linear regression between the annual temperature and the glacier area are -42.47 $km^2/°C$ (Zone I), -1669.57 $km^2/°C$ (Zone II), -1531.32 $km^2/°C$ (Zone III), -196.12 $km^2/°C$ (Zone VI), -34.65 $km^2/°C$ (Zone V), -5762.01 $km^2/°C$ (Zone VI), -277.9 $km^2/°C$ (Zone VII), and -9341.39 $km^2/°C$ (Zone VIII). Thus, across all the sub-zones, there is a negative correlation between glacier area changes and annual temperature increases. To avoid this confusion, a detailed table containing these values was added in the supplement. (Please see Table S1)

24. Lines 361-362: See previous comment. A visual interpretation of Figure 9b (annual) suggests that only zones III, V, VI and VIII have confidence intervals in excess of 0 on the y-axis, and that

therefore only these 4 zones can be confidently interpreted as exhibiting a positive relationship between glacier area and total precipitation at an annual scale. Equally, if the confidence intervals for zone II also cross the y-axis and it cannot be said with confidence that the relationship is negative. I think it would be beneficial for the authors to re-write this entire paragraph with greater consideration given to their confidence intervals, particularly where they cross 0 on the y-axis.

Response: Thank you for your kind suggestion and comment. In this revision, the detailed table presenting the regression results was added in the supplement. In Table S2, the regression coefficients, the 95% confidence intervals, standard error, t-values, p-values, and a summary of model diagnostics ($R^2$, Adjusted $R^2$, and RSE) were displayed. As such, the impacts of precipitation on the glacier area change in each sub-zone was reanalyzed. (Please see Line 402-411 and Table S2)

25. Line 363: Should 'zones III, V, and VIII' read 'zones III, VI and VIII'? It also looks like zone VI could be added to this list.

Response: Thank you for your comment. The correlation coefficient of the annual precipitation and the glacier area in Zone VI is 121 $km^2$/mm. However, this coefficient of determination ($R^2$) is only 0.19. Thus, this restraining impact of annual precipitation on the glacier area is not evident in Zone VI. To avoid this confusion, this explanation was added in this revision. (Please see Line 405-407)

26. Lines 363-365 and Figure 9b: Units are inconsistent here, the text states $km^2$/mm, but the y-axis on Figure 9b states $km^2$/m. The abstract also states $km^2$/mm.

Response: Thank you for your kind suggestion. The unit in Figure 9(b) is $10^3$ $km^2$/m, which is also $km^2$/mm. To avoid this confusion, the unit of Figure 9(b) was revised as $km^2$/mm. (Please see Figure 10b)

27. Line 383: Which earlier study; this study or Huang et al. 2021?

Response: Thank you for your comment. The study of Huang et al. (2021) is earlier. To avoid this confusion, this sentence was revised as follows: Huang et al. (2021) utilized the minimum NDSI value at each pixel for glacier mapping, which proved to be particularly effective in minimizing the influence of seasonal snow cover. (Please see Line 317-318)

28. Lines 405-406: For consistency, it would be useful to also include zone numbers when referring to locations in this sentence.

Response: Thank you for your kind suggestion and comment. In this revision, this sentence was revised as follows: The distinct regional variations in changes of glacier extent, with the most pronounced retreat observed in the Himalayas and the southeastern Tibetan Plateau (i.e., Zones II

and III), as shown in Fig. 4. (Please see Line 421-422)

29. Lines 409-411: See previous comment.

Response: Thank you for your kind suggestion and comment. In this, this sentence was revised as follows: The rising temperatures and declining precipitation in the southeastern Tibetan Plateau (i.e., Zone III), as depicted in Fig. 8, are likely contributing factors to the glacier retreat observed in this region. Moreover, a gradient of glacier shrinkage is observed, diminishing progressively from the Himalayas towards the continental interior (i.e., From Zone II to Zone V), indicating a less pronounced retreat in these regions. (Please see Line 424-428)

30. Lines 432-436: Given the scarcity of data points I think the authors should be cautious of giving too much weight to the data presented in Figure 11. For example, although zones I and III have lower median debris thickness and relatively high rates of glacier area loss, zone 8 has a similar level of debris thickness, but much lower rates of glacier area loss, and zone 4 has the highest median debris thickness, but still has the 3rd highest rate of glacier area loss. This paragraph could benefit from a discussion of the differences between the regional glacier area losses presented here (for debris-free ice) and any studies of regional glacier area losses for debris-covered glaciers (should the data be available).

Response: Thank you for your kind suggestion and comment. The data on glacier area losses for debris-covered glaciers are lacking in previous studies; thus, the comparisons of the glacier area loss between the debris-free glaciers and debris-covered glaciers were not discussed in this study. To avoid such confusion, the discussion on the influence of debris thickness on the glacier retreat was weakened and shortened. (Please see Line 458-460)

31. Line 451: Clarify that the 'total glacier area' is the debris-free total glacier area.

Response: Thank you for your comment. In this revision, this sentence was revised as follows: The total debris-free glacier area decreased from $94.59 \times 10^3$ km$^2$ to $61.16 \times 10^3$ km$^2$, with an average annual retreat rate of $1.08 \pm 0.28 \times 10^3$ km$^2$. (Please see Line 491-492)

32. Line 457-458: I think this sentence should be rephrased because the total glacier area in these regions has typically reduced over the duration of the study (see trends on Figure 4), rather than advanced. Given the overall reduction in glacier area and the correlations presented in Fig. 9b it may be more appropriate to state that annual and summer precipitation may have helped to mitigate reductions in glacier area in the Karakorum.

Response: Thank you for your kind suggestion and comment. In this revision, this sentence was revised as follows: A slight increase in precipitation is mainly observed in the southwest Himalayas

and the Karakoram, which may have helped to mitigate reductions in the glacier area in these regions. (Please see Line 497-498)

33. Lines 461-464: I think this conclusion could also be revised with regard to the earlier comment about confidence intervals (see comment for Lines 361-362).

Response: Thank you for your kind suggestion and comment. In this revision, the conclusion was revised considering the confidence interval of the regression coefficient. (Please see Line 503-504)